# Bridging the Gap to Real-World Language-Grounded Visual Concept Learning

**Whie Jung**    **Semin Kim**    **Junee Kim**    **Seunghoon Hong**
School of Computing, KAIST
{whieya, seminkim, kje0312, seunghoon.hong}@kaist.ac.kr

## Abstract

Human intelligence effortlessly interprets visual scenes along a rich spectrum of semantic dimensions. However, existing approaches to language-grounded visual concept learning are limited to a few predefined primitive axes, such as color and shape, and are typically explored in synthetic datasets. In this work, we propose a scalable framework that adaptively identifies image-related concept axes and grounds visual concepts along these axes in real-world scenes. Leveraging a pretrained vision-language model and our universal prompting strategy, our framework identifies a diverse image-related axes without any prior knowledge. Our universal concept encoder adaptively binds visual features to the discovered axes without introducing additional model parameters for each concept. To ground visual concepts along the discovered axes, we optimize a compositional anchoring objective, which ensures that each axis can be independently manipulated without affecting others. We demonstrate the effectiveness of our framework on subsets of ImageNet, CelebA-HQ, and AFHQ, showcasing superior editing capabilities across diverse real-world concepts that are too varied to be manually predefined. Our method also exhibits strong compositional generalization, outperforming existing visual concept learning and text-based editing methods. The code is available at https://github.com/whieya/Language-grounded-VCL.

## 1 Introduction

Perceiving the world through visual concepts such as color, shape, and texture, as human intelligence does, has long been a goal in computer vision. Representing an image as a composition of these concepts not only improves compositional generalization [8, 25, 36, 37], but also offers interpretable explanations [14] and enhances visual reasoning tasks [9, 38]. Early work primarily used discrete language descriptors, ranging from object labels in classification and detection [7, 15, 20, 31] to sentence-level captions [1, 34]. A recent method [16] shows that continuous concept embeddings, grounded along language-informed axes, can capture subtle visual nuances, *e.g.*, slight color variations, beyond the reach of purely text-based approaches. Thanks to the visual nuances embedded in continuous representations, this method enables the transfer of subtle, image-dependent details in downstream tasks such as image-editing tasks, where discrete text descriptor-based approaches [3, 24] often struggle due to limited linguistic expressiveness.

Despite this promise, extending the recent approach [16] to learn diverse visual concepts in real-world scenes remains underexplored. A central challenge is the reliance on *predefined* concept axes, such as color or shape, for visual grounding, which fails to capture the rich diversity of real images and limits extension to datasets where relevant factors are unknown in advance. Moreover, since each image consists of a wide variety of concept axes, relying on a specialized concept encoder for every axis quickly becomes infeasible, substantially increasing model complexity. Constraining each concept embedding to contain information relevant only to a specific concept axis presents another significant challenge. Although directly matching concept embeddings to textual descriptors—already a disentangled term in nature—offers a simple remedy for disentanglement [16], it compromises instance-specific details, as textual descriptors are image-agnostic.

39th Conference on Neural Information Processing Systems (NeurIPS 2025).

In this work, we take a step toward a scalable approach for visual concept learning in real-world scenes. We leverage a pretrained vision-language model (VLM) to adaptively identify image-related axes, replacing fixed predefined ones. Using a universal prompt design, we guide the VLM to identify diverse image-related axes without relying on prior knowledge. Our universal concept encoder then binds visual features to these discovered axes within a single unified architecture. To ensure that the discovered axes remain disentangled while preserving image-specific details, we introduce a compositional anchoring objective that constrains changes within each axis so that they only affect the corresponding axis in the generated images. We demonstrate that our scalable framework can capture diverse real-world concepts and enable novel compositions of visual concepts.

In summary, our contributions are as follows:

1. We introduce a scalable framework that grounds visual concepts along diverse, language-specified axes in real-world images.

2. We propose adaptively identifying image-related axes with a pretrained VLM and designing a universal concept encoder that binds visual features to these axes.

3. We design a novel objective for disentangling discovered concept axes in real-world scenes.

4. We evaluate our framework on real-world concept editing tasks, showing superior editing capabilities and compositional generalization compared to language-informed visual concept learning methods and text-based editing methods.

## 2   Problem Setup

Our goal is to develop a scalable framework for extracting visual concepts grounded along image-related linguistic axes in real-world images. To this end, we first outline a general formulation of language-grounded visual concept learning and identify the key challenges in scaling to real-world scenarios. Given an input image $\mathbf{x} \in \mathbb{R}^{H \times W \times C}$, the objective is to extract a set of concept representations $Z = \{\mathbf{z}_1, \ldots, \mathbf{z}_K\}$, where $\mathbf{z}_i \in \mathbb{R}^D$ encodes visual concepts relevant to concept axis $y_i$. To define interpretable axes among infinitely many concept axes in real-world images, we define each concept axis $y_i$ with natural languages, *e.g.*, age, gender, and expression. Then the goal is to learn a set of concept encoders $E_{\theta_i}$ mapping $\mathbf{x}$ to visual concepts $\mathbf{z}_i$ corresponding to each concept axis $y_i$. A typical approach to train such encoders is training jointly with a decoder $D$ with an auto-encoding objective. The decoder $D$ is often replaced by a frozen pre-trained text-to-image (T2I) generative model [16] due to training efficiency and remarkable generation capabilities. Formally, the encoders are optimized with the denoising objective:

$$\mathcal{L}_{\text{Diff}}(\{\theta_i\}) = \mathbb{E}_{\epsilon,t} \left[ ||D(\mathbf{x}, t, \{E_{\theta_i}(\mathbf{x})\}) - \epsilon||_2^2 \right] \tag{1}$$

where $\epsilon \sim \mathcal{N}(\mathbf{0}, I)$ and $t \sim U(0, 1)$ denote noise and timestep, respectively.

Since Equation 1 does not guarantee the disentanglement of visual concepts along the concept axes, prior work [16] introduces additional regularization to ground each visual concept $\mathbf{z}_i$ to the text embeddings $\mathbf{v}_i$, which are obtained by querying the pretrained VLM [18] with predefined templates, *e.g.*, "what is the color of the object". We denote by $v_i$, *e.g.*, red or blue, the textual descriptions for each axis, and define $\mathbf{v}_i = T(v_i)$ as their embeddings, where $T$ is a pretrained text encoder.

### 2.1   Challenges and Desiderata

While prior work [16] demonstrated the extraction of primitive visual concepts, *e.g.*, color, shape, and style, primarily on simple synthetic datasets, extending this method to complex real-world scenes poses three key challenges. We briefly outline these challenges and desiderata in this section, and discuss how they are addressed in Section 3.

**Adaptive Concept Axes**   Concept axes for visual grounding should be determined adaptively for each image, since real-world images exhibit a vast diversity of attributes that cannot be covered by a fixed set of predefined axes. Rather than relying on predefined primitive axes, *i.e.*, color or shape, an adaptive mechanism is required to automatically identify relevant concept axes for each image.

**Scalable Encoder Architecture**   To support adaptive concept axes, the encoder architecture should be scalable. Implementing $E_\theta$ with a set of specialized concept encoders for each concept axis $y_i$ would incur a prohibitive number of model parameters, considering infinitely many potential concept axes in real-world scenes.

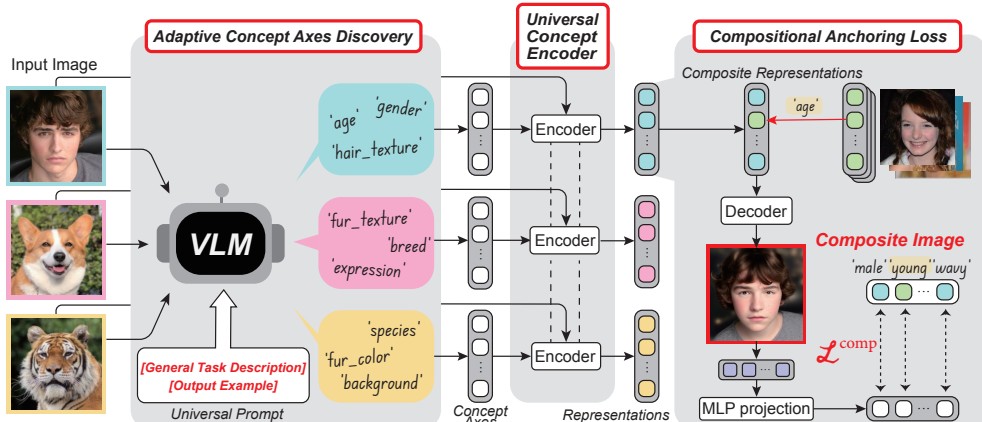

Figure 1: Overview of our method. Our framework first identifies image-related concept axes by leveraging VLM. We design an universal prompt that guides the VLM to find concept axes across different datasets. Given discovered axes for each image, our universal concept encoder binds visual features to those axes without introducing any specialized concept encoder for each axis. Finally, the encoded concept representations are regularized with a compositional anchoring loss to promote disentanglement between concept axes. Specifically, we randomly swap a concept representation with the one in an identical concept axis extracted from different images, and constrain composite images, rendered from randomly swapped representations, to be aligned with composite text descriptions.

**Concept Disentanglement** Given adaptive concept axes, each representation $z_i$ should capture only the semantics of its corresponding axis $y_i$, while preserving image-specific details. A straightforward solution is to align $z_i$ with the text embedding $v_i$ [16], since texts are already disentangled along concept axes in nature. However, since $v_i$ does not encode any instance-specific information, this alignment often leads to a suboptimal trade-off in $z_i$ between encoding visual nuisance and disentanglement of the concepts.

## 3 Approach

Based on the desiderata outlined in Section 2.1, we present a scalable framework for language-grounded visual concept learning (Figure 1). To extract concept axes adaptive to given images, we propose to leverage a pretrained VLM with our simple yet effective prompting strategy (Section 3.1). Given adaptive concept axes for each image, our universal concept encoder maps the image features to their corresponding visual concept embeddings (Section 3.2). We then train this encoder to disentangle visual concepts along the discovered axes by maximizing compositional anchoring of the representations (Section 3.3). Instead of directly aligning $z_i$ with the image-agnostic text embedding $T(v_i)$, compositional anchoring ensures that changes in $z_i$ affect only its corresponding concept axes $y_i$ in the generated image space $D(Z)$. Below, we describe each component in detail.

### 3.1 Adaptive Concept Axes Discovery

Given an image $\mathbf{x}$, we query a pretrained VLM with a prompt $\mathcal{P}$ to extract image-dependent concept axes $Y = \{y_1, \ldots, y_K\}$ and their corresponding textual descriptions $V = \{v_1, \ldots, v_K\}$. Note that $K$ varies per image, and the extracted descriptions $V$ will be used for visual grounding in Section 3.3. The prompt $\mathcal{P}$ should be universal, generalizing across arbitrary images and properly guiding the VLM to capture rich image-related concepts. To this end, we design a universal prompt with two key components: a **general task description** and an **output exemplar**. The general task description instructs the VLM to enumerate all visually relevant concept axes presented in a given image. On the other hand, the **output exemplar** demonstrates the desired granularity of axes by providing a specific instance. By specifying axes in the exemplar, VLM can be steered to find more detailed axes, *e.g.*, hair color, hair texture, and avoid overly coarse categories, *e.g.,* color, texture. Remarkably, a single exemplar is sufficient to steer the VLM to identify diverse image-related concepts beyond those provided in the exemplar and to generalize to new domains. For example, given an instance of a human face that includes the axis 'hair color', the VLM discovers unspecified attributes such as 'eye color' or 'lip color' for different human faces, and identifies analogous axes for animal images, *e.g.*, fur color. See Appendix A.4 for more details.

We also instruct the VLM to structure the output as a dictionary mapping each concept axis to a corresponding textual description, *e.g.* {'age': 'young', 'gender': 'male',...}. While prior work [16] employs the VLM to gather textual descriptions for a few predefined axes, they query the model separately for each predefined concept axis (e.g., "What is the color of the image?"). In contrast, our prompting strategy extracts all concept axes and corresponding textual descriptions in a single query, greatly enhancing efficiency and covering a broader range of potential axes beyond typical predefined categories. The complete prompt and outputs are provided in Appendix A.4. We find this universal prompt effectively captures diverse image-related concepts across multiple datasets, including novel concepts, *e.g.*, breed, eye color, and nose color, which were *not present* in the exemplar.

## 3.2 Universal Concept Encoder

In our framework, the concept encoder $E_\theta$ requires to encode visual concepts adaptive to image-related concept axes $Y$. Rather than defining specialized concept encoders for each concept, we construct $E_\theta$ to encode all concept representations $Z$ conditioned on a set of concept axes $Y$, *i.e.*, $Z = E_\theta(\mathbf{x}, Y)$. The architecture for $E_\theta$ should support adaptive binding of visual features to given axes $Y$ and produce distinct concept representations within a single parameterized model. To this end, we adapt the Querying Transformer (Q-Former) [18], which was designed to extract visual features from a frozen vision encoder and align them with pretrained text embeddings. The Q-Former consists of a lightweight transformer with learnable queries to interact with visual features via a cross-attention module. In our adaptation, we replace the learnable queries with the text embeddings $T(\{y_i\})$ of each axis encoded from a pretrained text encoder $T$. Initial queries $T(y_i)$ are then updated in subsequent transformer layers by interacting with visual features through cross attention layers. This way, visual features can dynamically bind to arbitrary concept axes within a single architecture.

## 3.3 Disentanglement with Compositionality

To constrain $\mathbf{z}_i$ to encode only the information relevant to its axis $y_i$, we introduce a *compositional anchoring* objective that ensures modifying a concept along one axis alters the generated output only in that axis, leaving other attributes unchanged. We implement such variations by randomly swapping a subset of concept representations $Z' \subseteq Z$ with those drawn from the same axis of different images, producing composite representations $Z^c$. As discovered axes vary across images, we first search for candidate images within each batch that share the same axis $y_i$, and then randomly swap their corresponding $\mathbf{z}_i$ among these candidates. When each representation $\mathbf{z}^i$ is disentangled along $y_i$, the composite image $\mathbf{x}^c = D(Z^c)$ should change only the swapped attributes, leaving others unchanged. Since ground-truth images for such a composition are generally unavailable, we instead measure alignment between the composite image $\mathbf{x}^c$ and composed textual descriptions set $V^c$, constructed by taking the corresponding descriptions from each swapped axis.

We quantify this alignment using a lightweight regression network $g_\phi$ that predicts the textual descriptions of a given image. Instead of constructing $g_\phi$ with an additional image encoder, we reuse $E_\theta$ to encode $\mathbf{x}^c$ back into concept representations, and a lightweight regression network $g_\phi$ predicts each attribute on top of the representations. Note that $g_\phi$ is shared across the axes. Formally, let $\hat{Z}^c = \{\hat{\mathbf{z}}_i^c\}_{i=1}^K = E_{\theta'}(\mathbf{x}^c, Y)$ be re-encoded concept representations from composite image $Z^c$, where $E_{\theta'}$ is a fixed copy of $E_\theta$. Then, the compositional anchoring objective is defined as:

$$\mathcal{L}_{\text{Comp}}(\theta) = \sum_{i=1}^K d\Big(g_\phi\big(\hat{\mathbf{z}}_i^c\big), \mathbf{v}_i^c\Big), \tag{2}$$

where $d(\cdot, \cdot)$ is a cosine distance and $\mathbf{v}_i^c$ is a text embedding for axis $y_i$ in $V^c$. Note that this objective only updates $\theta$ by propagating the gradient through $\mathbf{x}^c$ and prevents updating $g_\phi$ and $E_{\theta'}$ to avoid corruption from out-of-distribution samples of $D(\mathbf{z^c})$. For $g_\phi$, we simply train it by predicting the text embeddings $\mathbf{v}_i$ from non-swapped concept representations $\mathbf{z}_i$:

$$\mathcal{L}_{\text{Reg}}(\theta, \phi) = \sum_{i=1}^K d\Big(g_\phi\big(\mathbf{z}_i\big), \mathbf{v}_i\Big), \tag{3}$$

It is worth noting that our objectives do not force $\mathbf{z}_i = \mathbf{v}_i$, which compromises instance-specific details in $\mathbf{z}_i$. Instead, disentanglement is encouraged by verifying that each axis remains independent in the generated output. As a result, our objective ensures concept disentanglement while retaining instance-dependent information, particularly crucial in complex real-world scenarios.

### 3.4 Learning objectives

In this section, we summarize our complete framework and learning objectives. Given an image $\mathbf{x}$, the VLM extracts a set of image-related axes $Y$. The universal concept encoder $E_\theta$ is then trained with an autoencoding objective, while the pretrained decoder $D$ remains fixed. To encourage disentanglement among axes, we randomly swap each concept representation $\mathbf{z}_i$ with another from the same axis in the batch, and measure the alignment of composite image $\mathbf{x}^c = D(Z^c)$ with its corresponding text embeddings $\mathbf{v}^c$ through a lightweight regression network $g_\phi$. The overall objective is:

$$\mathcal{L}_{\text{Total}}(\theta, \phi) = \mathcal{L}_{\text{Diff}}(\theta) + \lambda_{\text{Comp}}\mathcal{L}_{\text{Comp}}(\theta) + \lambda_{\text{Reg}}\mathcal{L}_{\text{Reg}}(\theta, \phi), \tag{4}$$

where $\lambda_{\text{Comp}}$ and $\lambda_{\text{Reg}}$ are hyper-parameters controlling the importance of each term.

## 4 Related Work

**Visual Concept Learning**   As language offers a human-interpretable interface, grounding visual concepts in natural language has long been a central goal in computer vision. Early efforts primarily aligned images with word-level annotations or object labels, supporting classification and detection tasks [15, 20, 31]. Extensions to neuro-symbolic frameworks [19, 22], integrating with visual concept learning, further advanced visual reasoning. Such language-based grounding not only enhanced interpretability [14], but also improved downstream performance on vision tasks [17, 18]. However, discrete text descriptors inherently limit the representational capacity to a fixed vocabulary. To address this, LIVCL [16] followed Textual Inversion-based approaches [10] by optimizing concept encoders with a pretrained T2I model to reconstruct the given images. While promising, the scope of the work was limited to a few predefined primitive concept axes.

**Representation Learning with Compositionality**   Another line of research explores *object-centric learning* to uncover generative factors. Recent methods [12, 35] compose latent representations from multiple images similar to our framework, but under more restrictive assumptions. For instance, L2C [12] randomly mixes object representations to produce composite images and maximizes the likelihood of these composites to learn object-centric representations. Wiedemer et al. [35] provides a theoretical analysis for compositional generalization and measures compositional consistency through a cyclic distance between latent representations and their reconstructions. However, this formulation relies on architectural constraints such as additive decoders, making them effective mainly on synthetic or low-complexity data. Without additive decoders, it can lead to a trivial solution where a single latent encodes all. In contrast, our approach employs a pretrained T2I model without imposing additional constraints, addressing real-world scenes with diverse concept axes. Instead of focusing on isolated objects, our compositional consistency objective promotes disentanglement among discovered concept axes and does not require a specialized decoder structure.

## 5 Experiment

### 5.1 Experiment Setup

**Implementation Details**   We leverage InternVL [4] for an open-sourced VLM. To handle complex real-world images, we employ DINO-v2 [26] to encode the image into visual features followed by our concept encoder, and employ Stable Diffusion-based T2I decoder [30] finetuned at $256\times256$ resolution. When generating composite images with the T2I decoder, we iteratively decode for 10 steps using DDIM [33]. Since propagating gradients through all these decoding steps is computationally expensive, we follow [6, 27] and truncate gradients at the last few decoding iterations. Lastly, we employ 2-layer MLPs for $g_\phi$. See Appendix A.3 for additional implementation details.

**Dataset**   We validate our framework on complex and unstructured real-world data, where each image contains a diverse set of conceptual axes that is infeasible to manually predefine these axes to cover all possible variations within the data. To this end, we first conduct experiments on a subset of the ImageNet dataset. We randomly sampled 20 classes from ImageNet (referred to as ImageNet-S20), covering categories such as animals (*e.g.*, tree frog, American black bear, sulphur butterfly, giant panda), everyday objects (*e.g.*, padlock, grand piano, motor scooter), and scenes (*e.g.*, boathouse, water tower), yielding approximately 28k training images ($\sim$1.4k images per class). This dataset presents a challenging scenario as each class contains diverse, image-specific visual concepts that are often not shared by other classes. Given the infeasibility of manually defining all the concept axes in ImageNet-S20 for prior visual concept learning methods [16], we additionally compare our

Table 1: Comparisons on visual concept editing task. Our method consistently outperforms recent text-based editing methods [3, 11, 23, 24] and language-informed visual concept learning [16].

| Method | ImageNet-S20 | | CelebA-HQ | | AFHQ-Dog | | AFHQ-Cat | |
|---|---|---|---|---|---|---|---|---|
| | CLIP (↑) | BLIP (↑) | CLIP (↑) | BLIP (↑) | CLIP (↑) | BLIP (↑) | CLIP (↑) | BLIP (↑) |
| SDEdit [23] | 0.195 | 0.381 | 0.200 | 0.447 | 0.250 | 0.493 | 0.257 | 0.474 |
| InstructPix2Pix [3] | 0.198 | 0.383 | 0.202 | 0.425 | 0.230 | 0.467 | 0.246 | 0.471 |
| NullText Inversion [24] | 0.189 | 0.341 | 0.193 | 0.422 | 0.251 | 0.489 | 0.255 | 0.476 |
| DDPM-Inversion [11] | 0.243 | 0.467 | 0.220 | 0.483 | 0.266 | 0.516 | 0.266 | 0.494 |
| LIVCL [16] | - | - | 0.226 | 0.469 | 0.270 | 0.518 | 0.268 | 0.480 |
| Ours | **0.251** | **0.474** | **0.239** | **0.496** | **0.272** | **0.535** | **0.271** | **0.514** |

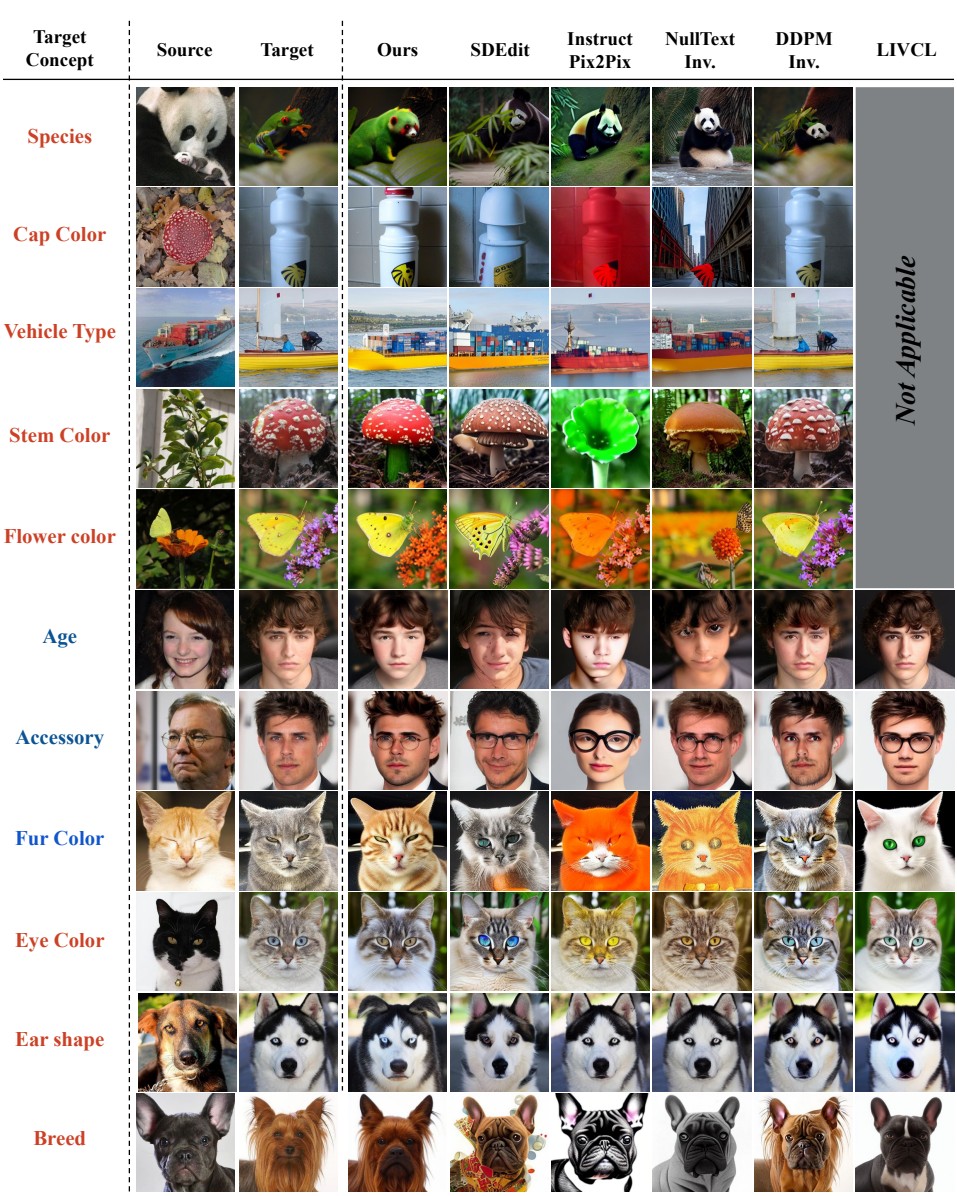

Figure 2: Qualitative results on ImageNet-S20, CelebA-HQ and AFHQ datasets. Our framework grounds visual concepts to diverse concept axes in real-world images. Note that **red concepts** are not provided by our prompt but rather adaptively discovered by VLM. Since it's infeasible to predefine all axes covering the whole dataset like ImageNet-S20, LIVCL was not applicable for ImageNet-S20.

approach with [16] using relatively controlled datasets with diverse concept axes, such as CelebA-HQ [13], AFHQ-Dog, and AFHQ-Cat [5]. We collect the frequently observed axes discovered by our method per dataset and use them to train the baseline [16] All images are resized to 256×256 for our experiments. For training and validation, we use the following splits: 28k/0.6k images for ImageNet-S20, 27k/3k for CelebA-HQ, and around 5k/0.5k for AFHQ-Dog and AFHQ-Cat.

**Evaluation Protocol**   To evaluate whether the concept representations faithfully capture their associated semantics and are disentangled from other axes, we perform a *visual concept editing* task. In this task, we select source and target images and identify the concept axes to be edited. The objective is to transfer a visual concept from the source to the target image without affecting other attributes. For evaluation, we use the top-50 and top-10 most frequently discovered axes per dataset for ImageNet-S20 and the remaining datasets, respectively, excluding axes that remain constant across the dataset, such as the subject type in CelebA-HQ, which is always human. For quantitative evaluation, we measure the CLIP-Score [29] and BLIP-Score [17] between the edited images and their corresponding swapped text descriptions $V^c$. Specifically, we construct text prompts $V^c$ such as "a photo of a cat with brown, fluffy, striped fur, against a black background," and evaluate the alignment with the images using CLIP and BLIP. Additionally, we conduct human evaluation, collecting responses from 10 participants per dataset via Prolific [28], following the procedure in Lee et al. [16]. Details on the human evaluation setup are provided in Appendix A.5.

**Baselines**   We compare our method to LIVCL [16], a recent visual concept learning approach that extracts concept representations along predefined primitive axes such as color, category, and style. As LIVCL explored in low resolution images, *e.g.*, 64×64 pixels, we replace its T2I decoder and pretrained image encoder with Stable Diffusion [30] and DINO- v2 [26], respectively. Since LIVCL requires predefined axes for training, we used the top-50/10 most frequent axes discovered by our method for ImageNet-S20 and the others, respectively. We also compare our method to four recent text-based image editing methods— SDEdit [23], InstructPix2Pix [3], Null-text Inversion [24], and DDPM-Inversion [11]. Although these baselines lack mechanisms for extracting visual concepts from source images, we instead edit the image with GT text descriptions given by the VLM. For each method, we used a prompt including target attributes to be changed, such as "a photo of a dog with brown fur" for editing. We used default hyper-parameters for text-based editing methods.

## 5.2   Main Results

**Quantitative Results**   We report quantitative comparison of our method to the baselines in Table 1. Our methods consistently outperform all baselines on all of the datasets by a clear margin. High CLIP and BLIP scores demonstrate the effectiveness of our method in capturing image-related visual concepts. A human evaluation in Table 2 provides a more direct assessment of reflecting subtle visual nuances. Since text-based editing methods are inherently independent of source

Table 2: Human evaluation results.

| Method | CelebA-HQ | AFHQ-Dog | AFHQ-Cat |
|---|---|---|---|
| SDEdit | 0.448 | 0.486 | 0.464 |
| InstructPix2Pix | 0.465 | 0.385 | 0.416 |
| NullText Inversion | 0.414 | 0.514 | 0.442 |
| DDPM-Inversion | 0.528 | 0.548 | 0.584 |
| LIVCL | 0.465 | 0.478 | 0.471 |
| Ours | **0.636** | **0.589** | **0.623** |

images and LIVCL struggles to encode image-dependent details due to its training objective, the performance gap becomes even more pronounced in the human evaluation. These results validate the effectiveness of our framework in visual grounding with diverse axes in real-world scenes.

**Qualitative Comparison**   Figure 2 presents the qualitative results on visual concept editing. Our method identified a diverse set of image-related axes and discovered novel concepts such as species, cap color, vehicle type, eye color, and breed, which were not specified in the prompt. It demonstrates that our universal prompt can generalize to unseen domains. Within the discovered axes, our method accurately alters each concept without affecting others. In contrast, LIVCL often fails to encode image-specific details, such as generating different glasses in the seventh row, last column, or disentangling from other axes like changing fur color and texture in the eighth row, last column. We attribute this to the inherent trade-off in LIVCL's objective between concept disentanglement and capturing image-dependent details. Thanks to our compositional anchoring objective, our method achieves both disentanglement along each axis and the preservation of image-specific details, *e.g.*, transferring similar glasses in the seventh row of third column. Text-based approaches also struggle with concept-wise manipulation, often modifying the global color of images (InstructPix2Pix and NullText-Inversion) or leaving them unchanged (SDEdit and DDPM-Inversion). Even when they transfer the correct attribute, they fail to capture the visual nuances of source attributes. For further visual inspection, please refer to additional qualitative results in Appendix A.6.

**Compositional Generalization**   Interestingly, our method demonstrates superior compositional generalization to unseen combinations of concepts compared to the baselines, as shown in Figure 3. In the figure, our method successfully generates novel compositions, such as a large frog with a panda's fur pattern, pandas with red eyes, or scooters floating on water, which do not exist in the real world. In

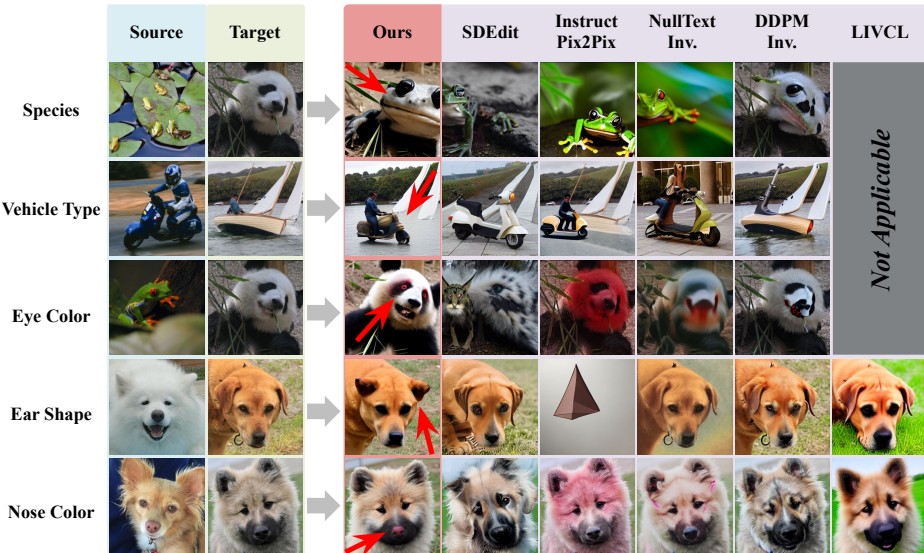

Figure 3: Compositional generalization to unseen concept combination. Given OOD combination of concepts such as frog with panda's fur pattern, only our method generates plausible results.

contrast, the baselines either alter multiple attributes simultaneously or change nothing at all, and fail to generate plausible generalizations. For instance, when modifying species or eye color, body colors are also changed as seen in the fifth and sixth columns of the first row, and the fifth column of the third row. Furthermore, while all baselines struggle with manipulating ear shapes or nose colors, which are strongly correlated with dog breed, our method shapes the ears of a Labrador into a triangle (third column of fourth row) and renders a dog's nose in pink (third column of fifth row). We conjecture that such compositional generalization arises from our compositional anchoring objective, which explicitly promotes random composite images to exhibit corresponding compositions of attributes.

**Composition From Multiple Images**    To further analyze the quality of extracted visual concepts, we consider the more challenging multi-image composition task. For each target image, we select $N$ source images and randomly sample unique concept axes from each source, *i.e.*, $N$ different axes. We then edit the target image along those axes to produce composite images. We conduct this task only on CelebA-HQ and AFHQ, as the high image diversity within ImageNet classes (*e.g.*, partial views, different viewpoints, or varying light conditions) often leads to cases where the concept axes are not consistently shared among the same class images, resulting in noisy evaluations. In contrast, CelebA-HQ and AFHQ have more controlled structures, making them better suited for this task.

Table 3 presents CLIP and BLIP scores for editing up to four axes. Our method again consistently outperforms all baselines. Moreover, all baselines suffer a clear drop in both metrics as $N$ increases, whereas our method shows only a marginal decrease. This robustness indicates that our concept representations are well disentangled and faithfully capture the correct semantics of the input images. Figure 4 shows qualitative results for $N = 3$. Composite images from our method are faithfully modified to reflect all of the source images' concepts. In contrast, the baseline models often omit or distort certain attributes. For example, all baselines fail to render the short hairstyle and blue earrings (first row), and some either drop the facial expression (InstructPix2Pix, DDPM Inversion) or misapply the hair color (LIVCL, SDEdit) in the composite outputs (second row).

**Visual Nuance Transfer**    In contrast to text-based editing methods, visual concept learning methods can capture visual nuances in the continuous representation space. Since LIVCL is not applicable to ImageNet-S20, we compare our methods to LIVCL on the CelebA-HQ and AFHQ datasets. Figure 5 highlights visual nuances captured in the concept representation of our method. In the figure, our method transfers subtle visual details such as detailed fur patterns, subtle differences in smiles, or hair color tones. In contrast, LIVCL struggles to correctly transfer these visual details, *e.g.*, the resulting image always exhibits the same expressions in Figure 5(b). It even fails to reconstruct the original images in Figure 5(c). This implies the suboptimal trade-off between concept disentanglement and image-dependent encoding induced by the objective in LIVCL. While pushing concept representations $\mathbf{z}_i$ closer to text embeddings $\mathbf{v}_i$, *i.e.*, $\mathbf{z}_i = \mathbf{v}_i$, can achieve disentanglement, it sacrifices visual information. In contrast, our compositional anchoring objective bypasses such

Table 3: Comparisons of visual concept editing. Our method outperforms recent text-based editing methods [3, 11, 23, 24] and language-informed visual concept learning [16].

| method | CelebA-HQ | | | | | | AFHQ-Dog | | | | | | AFHQ-Cat | | | | | |
|---|---|---|---|---|---|---|---|---|---|---|---|---|---|---|---|---|---|---|
| | CLIP | | | BLIP | | | CLIP | | | BLIP | | | CLIP | | | BLIP | | |
| | N=2 | N=3 | N=4 | N=2 | N=3 | N=4 | N=2 | N=3 | N=4 | N=2 | N=3 | N=4 | N=2 | N=3 | N=4 | N=2 | N=3 | N=4 |
| SDEdit | 0.203 | 0.202 | 0.204 | 0.443 | 0.440 | 0.439 | 0.251 | 0.254 | 0.255 | 0.493 | 0.496 | 0.494 | 0.257 | 0.254 | 0.254 | 0.466 | 0.456 | 0.443 |
| InstructPix2Pix | 0.201 | 0.199 | 0.197 | 0.416 | 0.413 | 0.411 | 0.225 | 0.224 | 0.221 | 0.456 | 0.458 | 0.453 | 0.248 | 0.250 | 0.249 | 0.462 | 0.463 | 0.450 |
| NullText Inv. | 0.206 | 0.199 | 0.194 | 0.428 | 0.420 | 0.417 | 0.247 | 0.250 | 0.250 | 0.479 | 0.482 | 0.480 | 0.256 | 0.257 | 0.258 | 0.471 | 0.469 | 0.468 |
| DDPM Inv. | 0.213 | 0.207 | 0.203 | 0.463 | 0.449 | 0.437 | 0.262 | 0.260 | 0.257 | 0.506 | 0.501 | 0.493 | 0.261 | 0.258 | 0.253 | 0.473 | 0.458 | 0.440 |
| LIVCL | 0.225 | 0.219 | 0.214 | 0.454 | 0.440 | 0.429 | 0.267 | 0.264 | 0.260 | 0.507 | 0.502 | 0.491 | 0.262 | 0.257 | 0.250 | 0.463 | 0.447 | 0.428 |
| Ours | **0.238** | **0.236** | **0.236** | **0.492** | **0.490** | **0.491** | **0.269** | **0.266** | **0.262** | **0.528** | **0.528** | **0.523** | **0.271** | **0.269** | **0.268** | **0.516** | **0.513** | **0.512** |

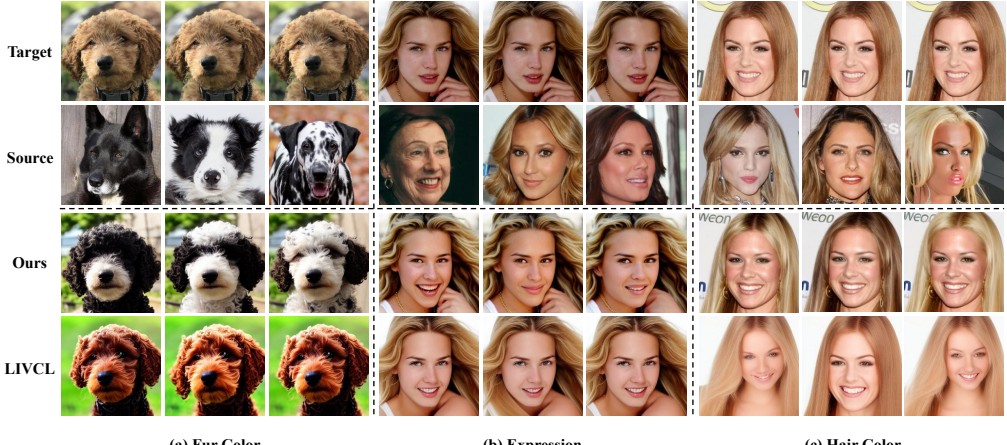

Figure 4: Composition of visual concepts from multiple images. Only our method accurately reflects all of the attributes of source images.

Figure 5: Examples of visual nuance transfer. Even when transferring the same attributes, *e.g.*, black and white fur or blonde hair, the outputs reflect subtle details of source images.

a trade-off and thereby achieves both disentanglement and rich image-dependent details within representations. More qualitative results on visual nuance transfer are provided in Appendix A.6.

## 5.3 Ablation Study

In this section, we conduct an ablation study on VLM choices, architectural design choices, and objective functions to examine the robustness and effectiveness of our choices. All of the experiments are evaluated on the visual concept editing task in the CelebA-HQ dataset.

**VLM choices** Since the discovery of concept axes in our method is directly affected by the quality of VLM outputs, we examine the robustness of our framework on two additional popular open-sourced VLMs (Qwen2.5-VL[2], Ovis2[21]), which have ranked highly on reasoning benchmarks.

Table 4: Ablation study on VLM choices.

| VLM choices | CLIP | BLIP |
|---|---|---|
| Qwen2.5-VL [2] | 23.72 | 48.64 |
| Ovis2 [21] | 23.48 | 48.35 |
| InternVL2-5 (Ours) | **23.88** | **49.58** |
| InternVL2-5 + 10% drop | 23.52 | 48.69 |
| InternVL2-5 + 20% drop | 23.65 | 48.61 |

Table 5: Ablation study on Architectural choices.

| Archiectural choices | | | Metrics | |
|---|---|---|---|---|
| Decoder | Vision Encoder | Concept Encoder | CLIP | BLIP |
| Frozen T2I | Dinov2 | UCE | **23.88** | 49.58 |
| LoRA-finetuned T2I | Dinov2 | UCE | 23.67 | **49.62** |
| Frozen T2I | CLIP | UCE | 22.03 | 46.21 |
| Frozen T2I | Dinov2 | Shared MLP | 21.63 | 46.8 |

Moreover, as it is difficult to directly control or quantify VLM performance, we instead control output quality by dropping partial axes (e.g., 10% and 20%) from the VLM outputs. It is a practical scenario as VLMs cannot always capture the complete axes for a given scene. Table 4 shows that our method is robust to both VLM choices and missing axes. We hypothesize that this is because even though some image-related axes can be missed in each example, those axes will eventually be repeatedly exposed across the dataset. Additionally, since our compositional anchoring loss encourages the compositionality of the concept representations, our framework might be internally trained for better compositional generalization, which improves adaptation with fewer samples. In fact, our method is capable of generating OOD samples (Figure 3). The robustness of our framework regarding the performance of VLMs suggests that it can scale to more complex real-world datasets, as VLMs do not always need to capture complete axes for every scene.

**Architectural choices**   Table 5 presents the ablation studies on architectural choices as follows: (1) Frozen T2I decoder versus LoRA-finetuned decoder, (2) choice of vision encoder (Dinov2 versus CLIP), and (3) universal concept encoder (UCE in the Table 5) versus shared MLP architectures. First, finetuning the decoder with LoRA does not affect overall performance. Large-scale pretrained T2I models have already learned expressive data priors on natural images, facilitating faster training of the generation model. Therefore, the frozen T2I model does not bottleneck our framework. Replacing the Dinov2 encoder with the CLIP encoder causes a significant performance drop, as CLIP is trained for text-alignment, making its discriminative properties inferior to those of recent self-supervised methods like Dinov2. Lastly, replacing our universal concept encoder with a shared MLP architecture, which is a naive version of an axis-agnostic encoder, also results in a severe drop. Specifically, the visual feature is mean-pooled into a vector, concatenated with axis embeddings, and passed through shared MLP layers to encode concept representations. To make this encoder generally work for diverse concept axes, we shared this MLP layer for all of the axes. This approach likely fails because the shared MLP treats each concept independently, blocking complex interactions between concepts. It clearly highlights the effectiveness of our universal concept encoder.

**Component-wise Contribution**   We conduct an ablation study on each component in our objective for concept disentanglement, *i.e.*, $g_\phi$ and $\mathcal{L}_{\text{Comp}}$, and report the results in Table 6. Without employing $g_\phi$ and instead directly regressing each concept representation $\mathbf{z}_i$ to $\mathbf{v}_i$ in Equation 2 and 3, we observe significant drops in both CLIP-Score and BLIP-Score. This result indicates the importance of $g_\phi$ in preventing a direct trade-off between disentanglement and encoding image-dependent

Table 6: Ablation study on our method. Both $\mathcal{L}_{\text{Comp}}$ and $g_\phi$ contribute to concept disentanglement.

| $\mathcal{L}_{\text{Comp}}$ | $g_\phi$ | CLIP ($\uparrow$) | BLIP ($\uparrow$) |
|---|---|---|---|
| ✓ | ✗ | 21.1 | 44.72 |
| ✗ | ✓ | 22.89 | 47.47 |
| ✓ | ✓ | **23.88** | **49.58** |

details. Furthermore, removing $\mathcal{L}_{\text{Comp}}$ also leads to suboptimal CLIP- and BLIP-Score. This is because minimizing Equation 3 only guarantees $\mathbf{z}_i$ to have information of $\mathbf{v}_i$ but does not prevent it from encoding entangled information related to other concept axes.

# 6   Conclusion

In this study, we present a scalable framework for grounding visual concepts along adaptive concept axes in real-world scenes. Our framework leverages a pretrained VLM and universal prompt design to adaptively identify diverse, image-related concept axes. A single, unified concept encoder then binds visual features to these axes, eliminating the need for separate per-concept encoders. To ensure each axis remains disentangled while preserving instance-level detail, we introduce a compositional anchoring loss. We randomly swap concept representations across images and regularize the resulting composite outputs to match their corresponding text descriptions. In the visual concept editing task on real-world datasets, our method consistently outperforms prior approaches in language-informed visual concept learning and recent text-based editing methods, demonstrating the effectiveness of our framework in learning adaptive visual concepts in real-world datasets. Also, our approach demonstrates successful transfer of subtle visual nuances and stronger compositional generalization.

**Acknowledgment**  This work was in part supported by the National Research Foundation of Korea (RS-2024-00351212 and RS-2024-00436165) and the Institute of Information & communications Technology Planning & Evaluation (IITP) (RS-2022-II220926, RS-2022-II220959, RS-2024-00509279, and RS-2019-II190075) funded by the Korea government (MSIT).

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

# A  Appendix

## A.1  Limitations and Future Work

In our work, as in most previous approaches, we cannot guarantee recovery of every ground-truth factor of variation. Some subtle or rare attributes may simply fall outside the axes we discover. In fact, perfectly capturing all underlying factors in a complex, real-world dataset is generally intractable. Nevertheless, our method still identifies diverse, meaningful concepts, and extending coverage to additional or more fine-grained factors remains an important direction for our future work. Moreover, our framework depends on the quality and scope of the pretrained vision language model (VLM), so it can only discover concepts the VLM recognizes. Fortunately, as VLMs are improving rapidly and our method is not restricted by a specific VLM, we can adopt stronger models as they become available.

## A.2  Broader Impact

Our approach can extract diverse visual concepts from images and reuse them to synthesize new content, which could pose privacy issues such as deepfake generation or unauthorized duplication of digital content.

## A.3  Additional Implementation Details

Table 7 summarizes hyper-parameters for model architectures and training used in our experiments. For baselines, we follow the default hyper-parameters recommended by the official codes. All baselines used DDIM inversion with guidance of 7.5 and 50 inference steps.

Table 7: Hyperparameters used in our experiments.

| | | |
|---|---|---|
| General | Batch Size | 32 |
| | Training Steps | 100k |
| | Learning Rate | 0.00003 |
| Concept Encoder | Layers | 4 |
| | Hidden Dim | 768 |
| | Number of Heads | 8 |
| Regression Network | Layers | 768 |
| | Input Dimension | 768 |
| | Hidden Dimensio | 768 |
| | Activation Function | ReLU |

## A.4 Prompt for Concept Axes Extraction

We provide the complete prompt and examples of the discovered concept axes per image in Figure 6 and Figure 7, respectively. As shown in Figure 7, our prompt successfully steers the VLM to identify diverse concept axes across different datasets, even when using only a single output exemplar of a human face.

**General Task Description**

<image>\n You are given an image containing any subject (e.g., person, animal, furniture, vehicle, etc.). Your task is to identify all relevant visual concept axes for the subject in a broad yet comprehensive way.Because the subject can vary widely, ensure your approach applies to humans, animals, vehicles, objects, and more.These axes should include all essential visible attributes of the subject with enough detail to plausibly reconstruct the image. Each axis should be fine-grained enough for distinct, specific attributes (for example, "hair_length" rather than "appearance", "fur_color" rather than "fur").
Each chosen axis (key) should map to a short, descriptive value.
Use concise yet precise terms. Avoid vague or overly brief descriptors like "medium" by replacing them with more specific language such as "long, past shoulders" or "short, tied back."\ Avoid repeating the same detail across multiple axes; keep attributes distinct to prevent overlap.
Return your final answer as a single JSON dictionary.

**Output Exemplar**

For example, if the subject is a human:
{"subject_type": "human", "age": "young adult","gender": "female","hair_color": "black","hair_length": "short, above ears","hair_texture": "straight","expression": "smiling","background": "white", accessory:"gold, dangling earrings"}

Output only the final JSON.

Figure 6: Our complete prompt consists of a general task description and output exemplar.

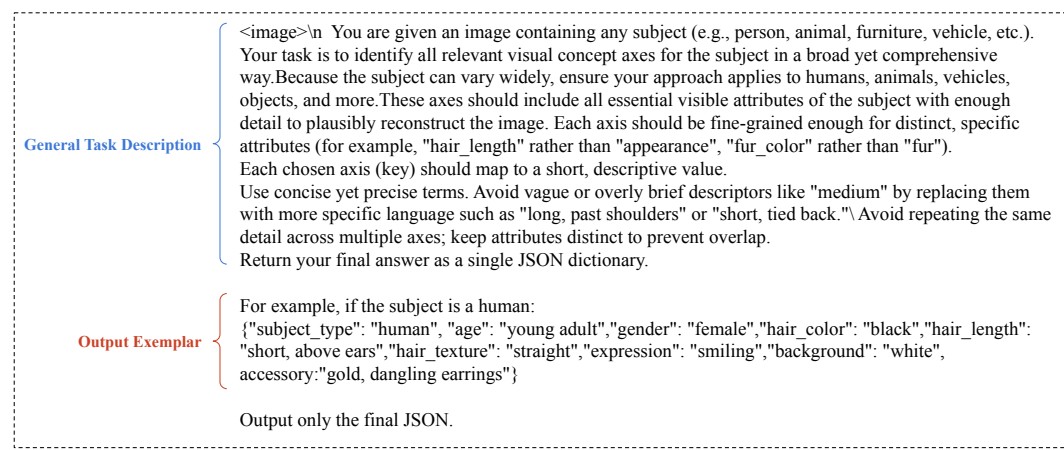

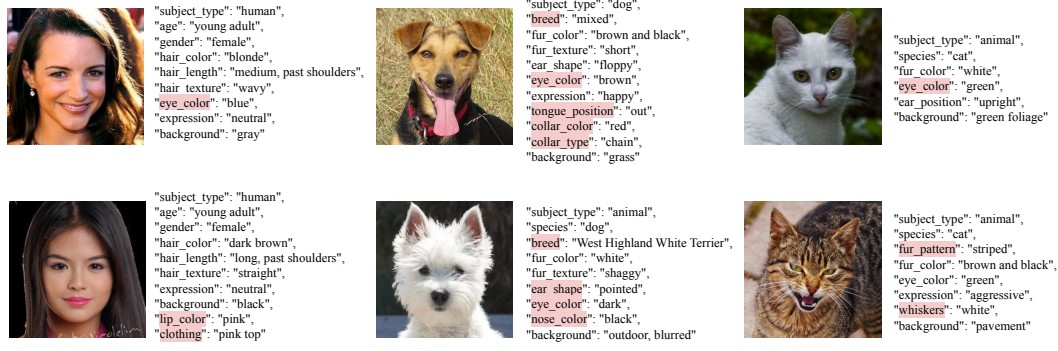

Figure 7: Examples of outputs from the VLM. Concept axes colored in red are unseen from the given exemplar.

### A.5 Human Evaluation

For human evaluation, we randomly select 10 pairs of images for each attribute. Then, we replace an attribute of one image with another one in each pair using each of the methods. We ensure that randomly selected attributes in each pair are different from each other so that the edited image is always recognizable. We collect 10 participants for each dataset (a total of 30) on Prolific [28] and provide a general guideline as in Figure 8 for the task. Our questionnaire (Figure 9) asks participants to rank the images that most closely adhere to the criteria provided in our guideline. Following [16], we used Borda score metrics [32] to differentiate the scores according to each ranking, and final scores are normalized to a 0-1 scale.

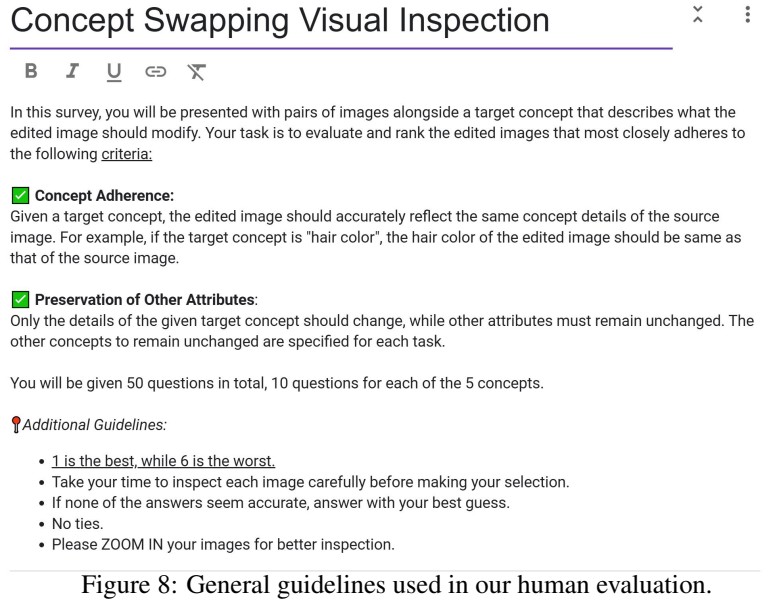

Figure 8: General guidelines used in our human evaluation.

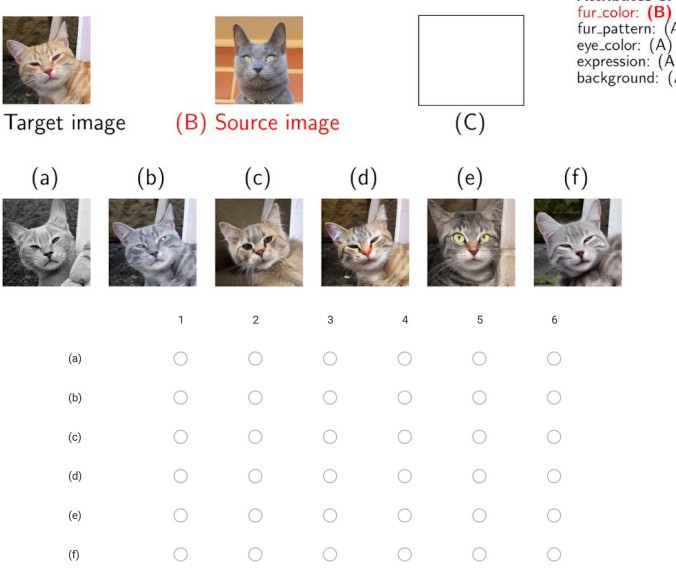

Figure 9: Questionnaires used in human evaluation.

## A.6 Additional Qualitative Results

### A.6.1 Additional Qualitative Comparisons on Visual Concept Editing

Figures 10–19 present additional qualitative results along diverse concept axes discovered in ImageNet-S20, CelebA-HQ, and AFHQ datasets. Across all axes, our method consistently outperforms the baselines. Whereas the baselines often fail to accurately capture and transfer the specified visual attributes, our approach reliably extracts the visual concept from the source and transfers it to the target image. Since LIVCL trains a set of separate encoders only for the top–10 frequent axes, it was unable to evaluate "lip color" in Figure 15 and "collar" in Figure 18, which are not among the top–10 most frequent concepts in the dataset, and we therefore mark those entries as N/A.

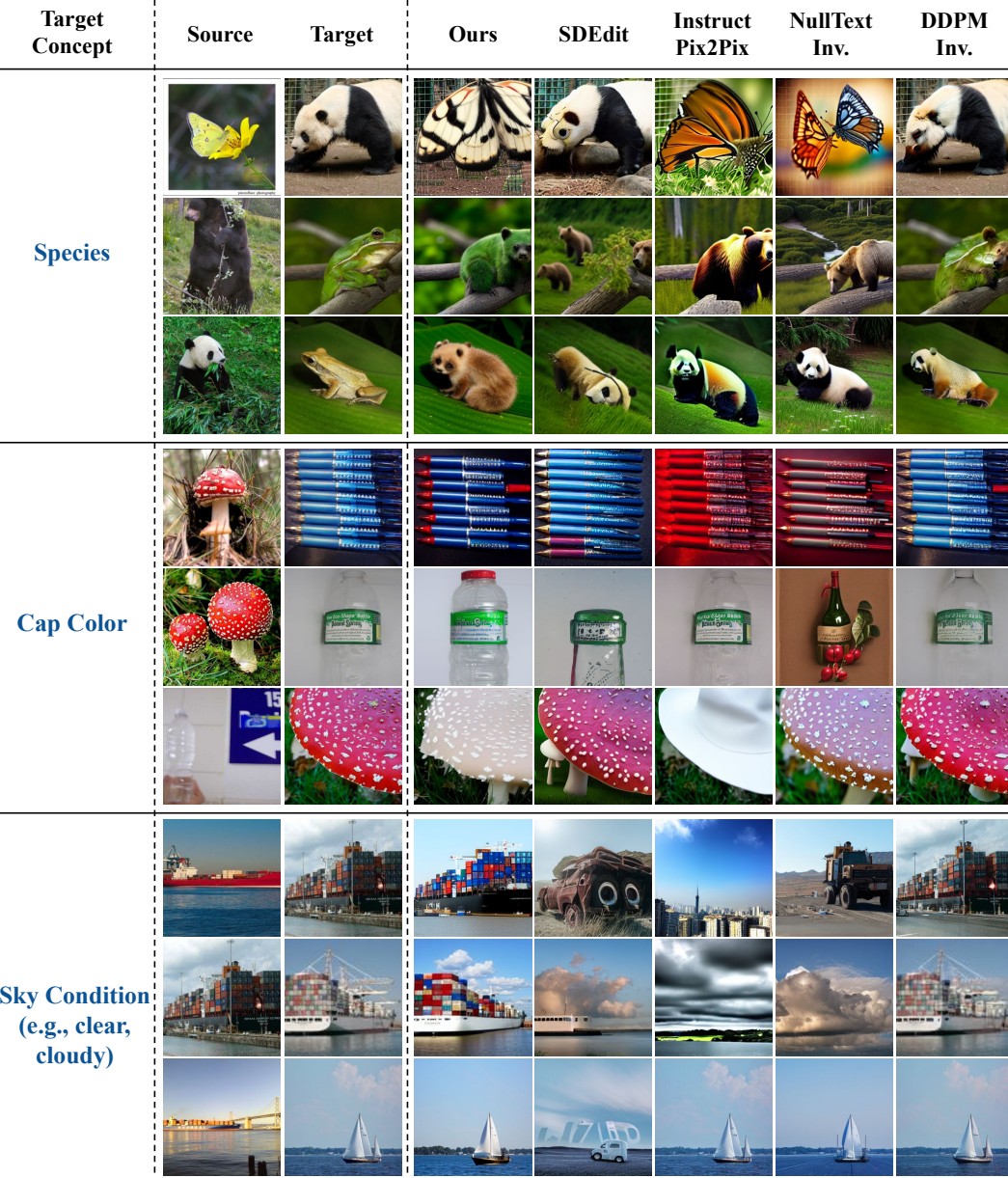

Figure 10: Additional qualitative comparison to baselines in ImageNet-S20

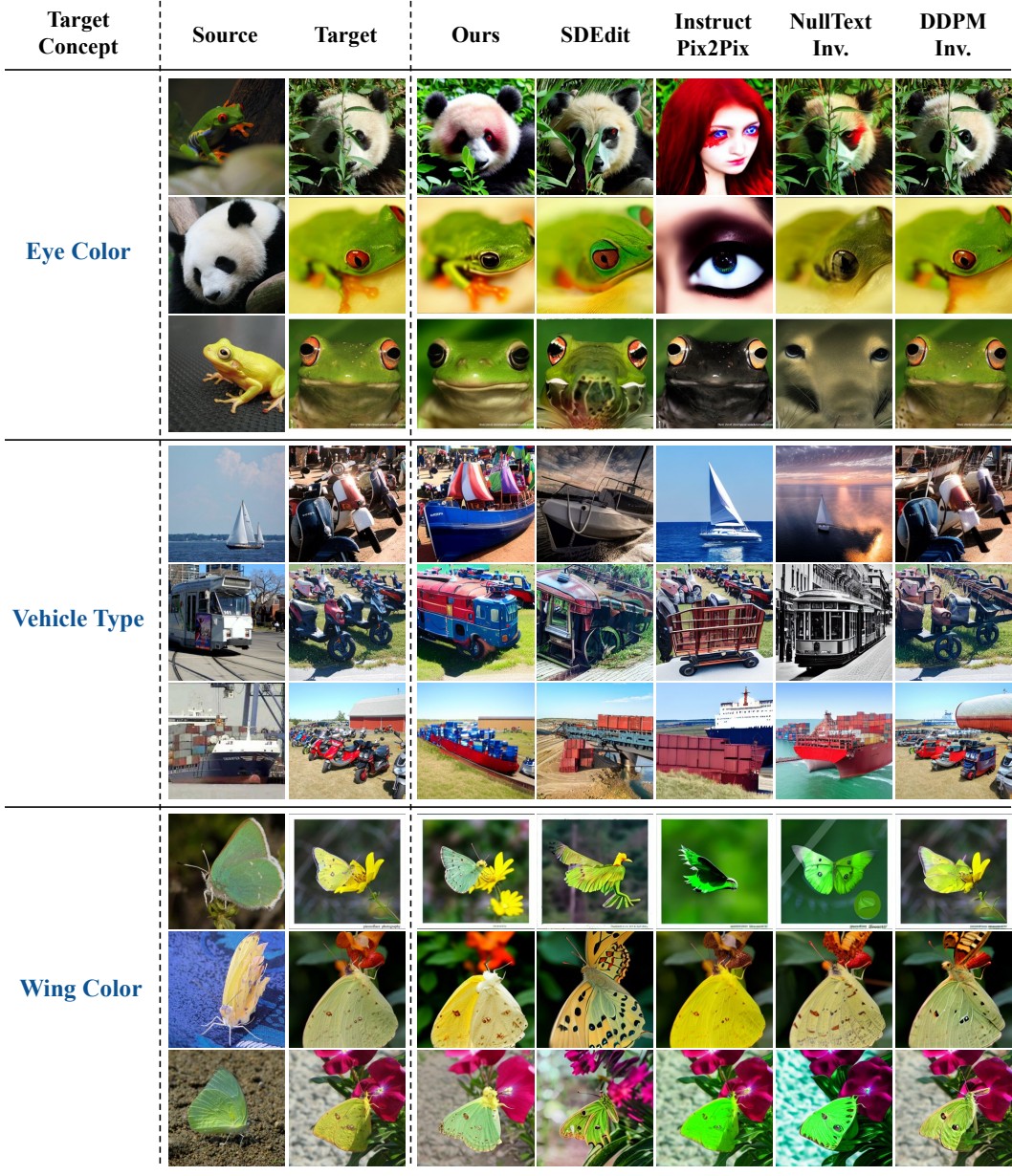

Figure 11: Additional qualitative comparison to baselines in ImageNet-S20

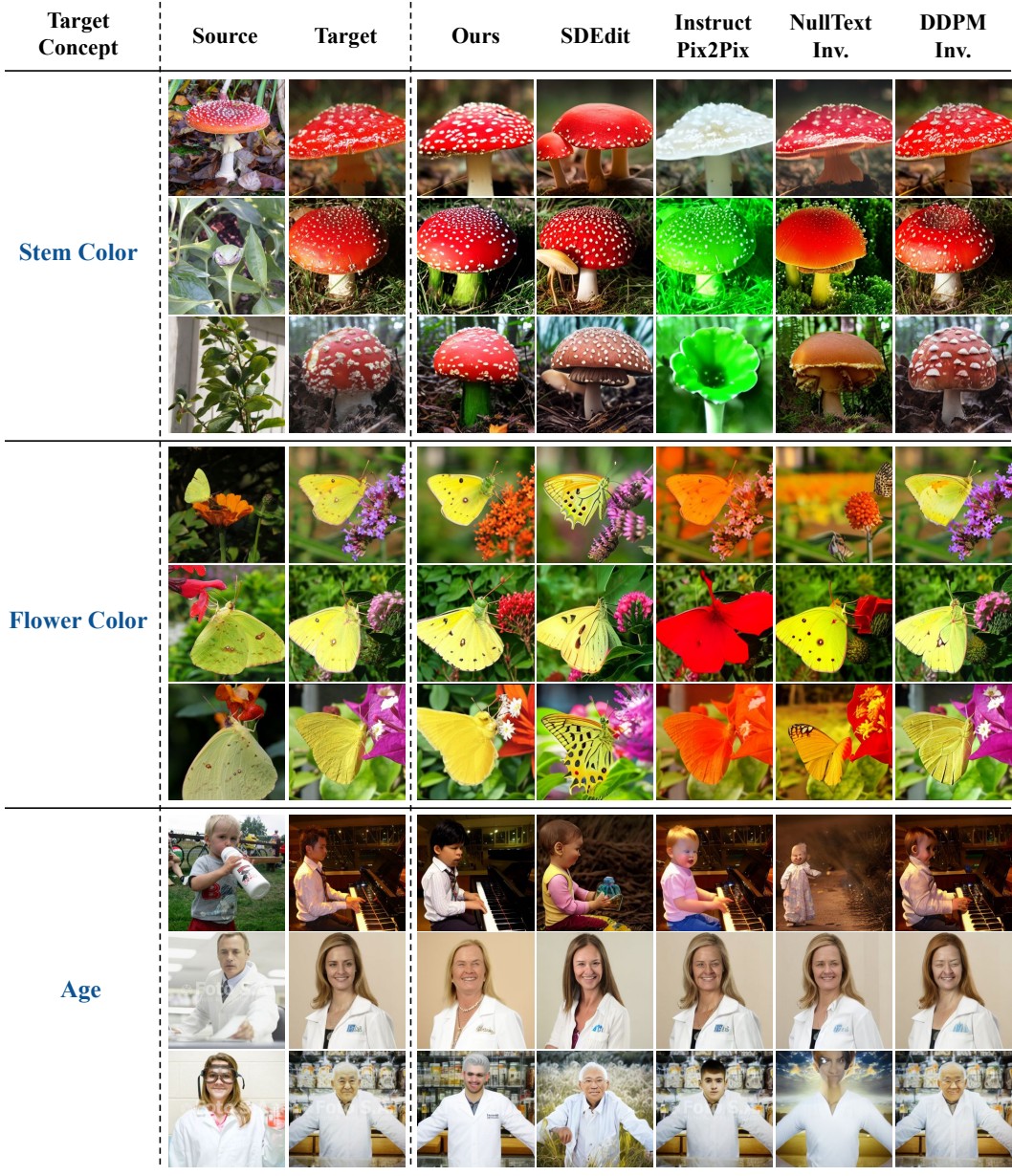

Figure 12: Additional qualitative comparison to baselines in ImageNet-S20

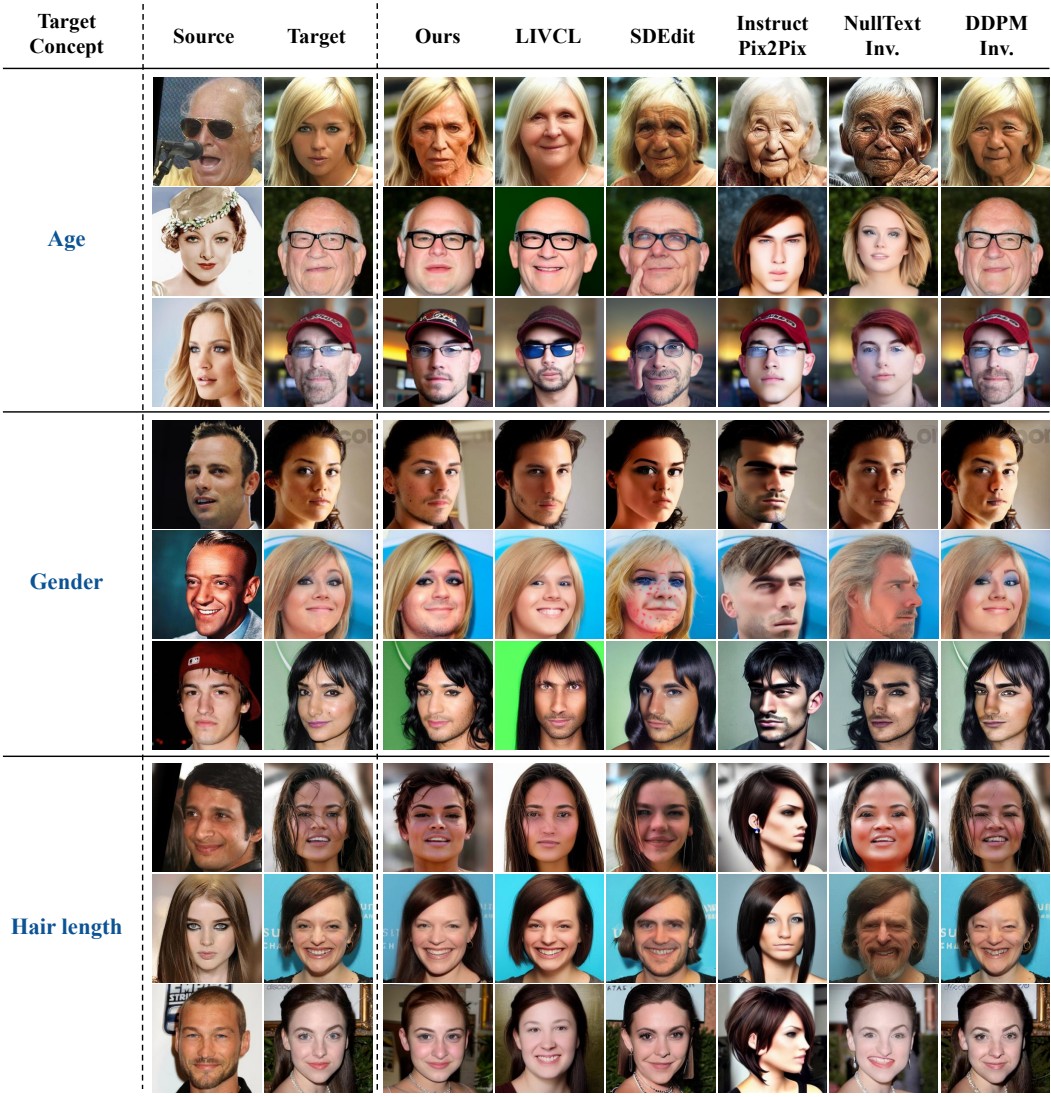

Figure 13: Additional qualitative comparison to baselines in CelebA-HQ

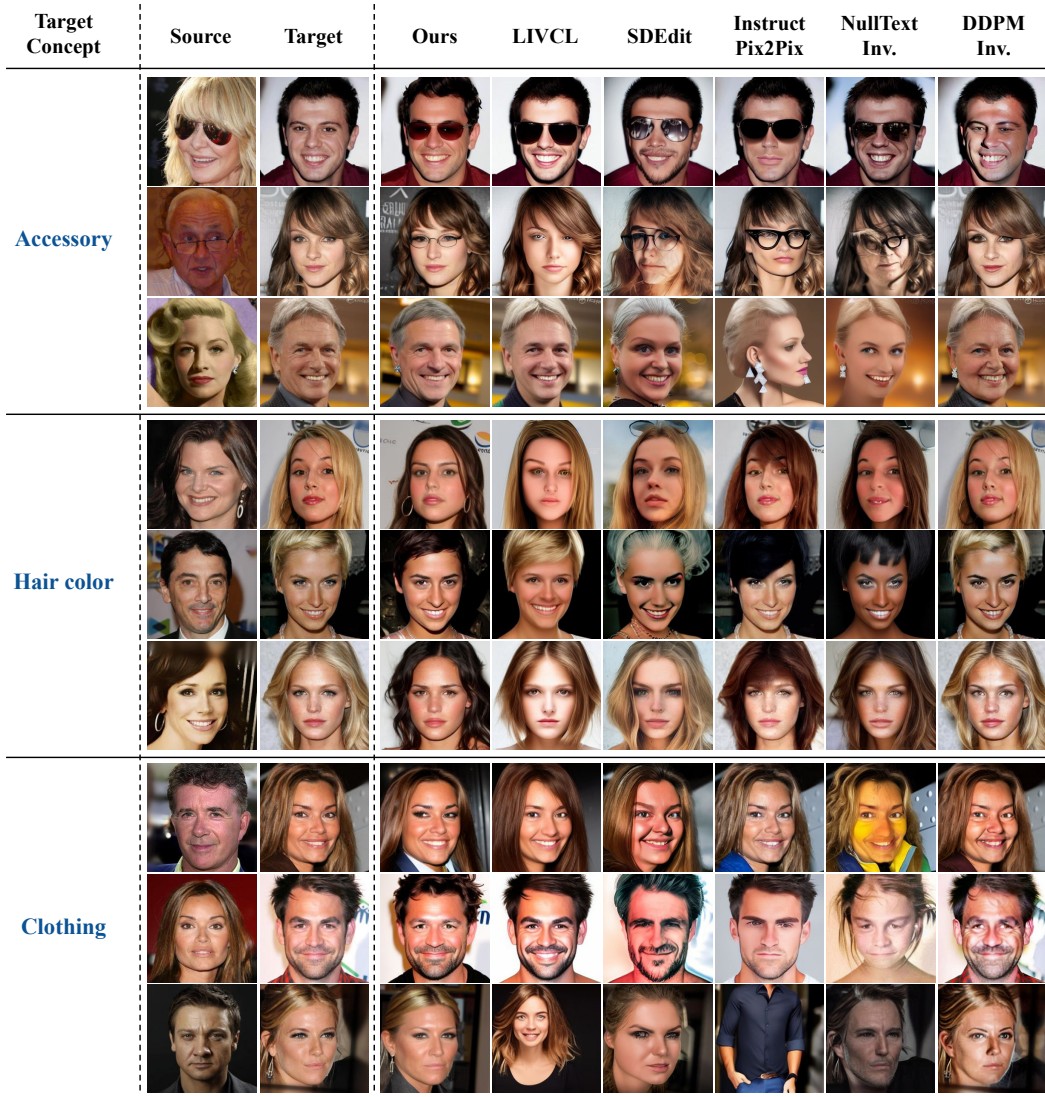

Figure 14: Additional qualitative comparison to baselines in CelebA-HQ

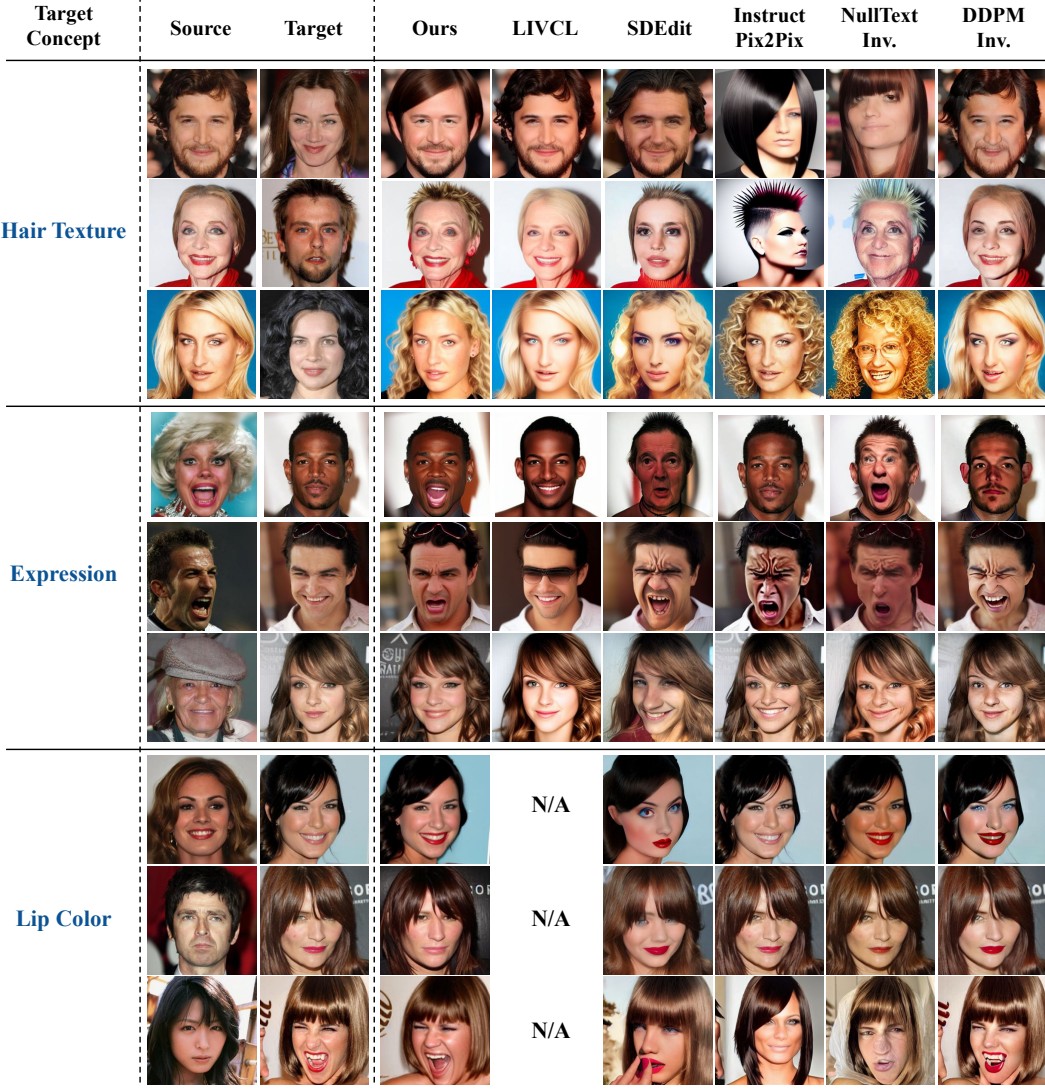

Figure 15: Additional qualitative comparison to baselines in CelebA-HQ

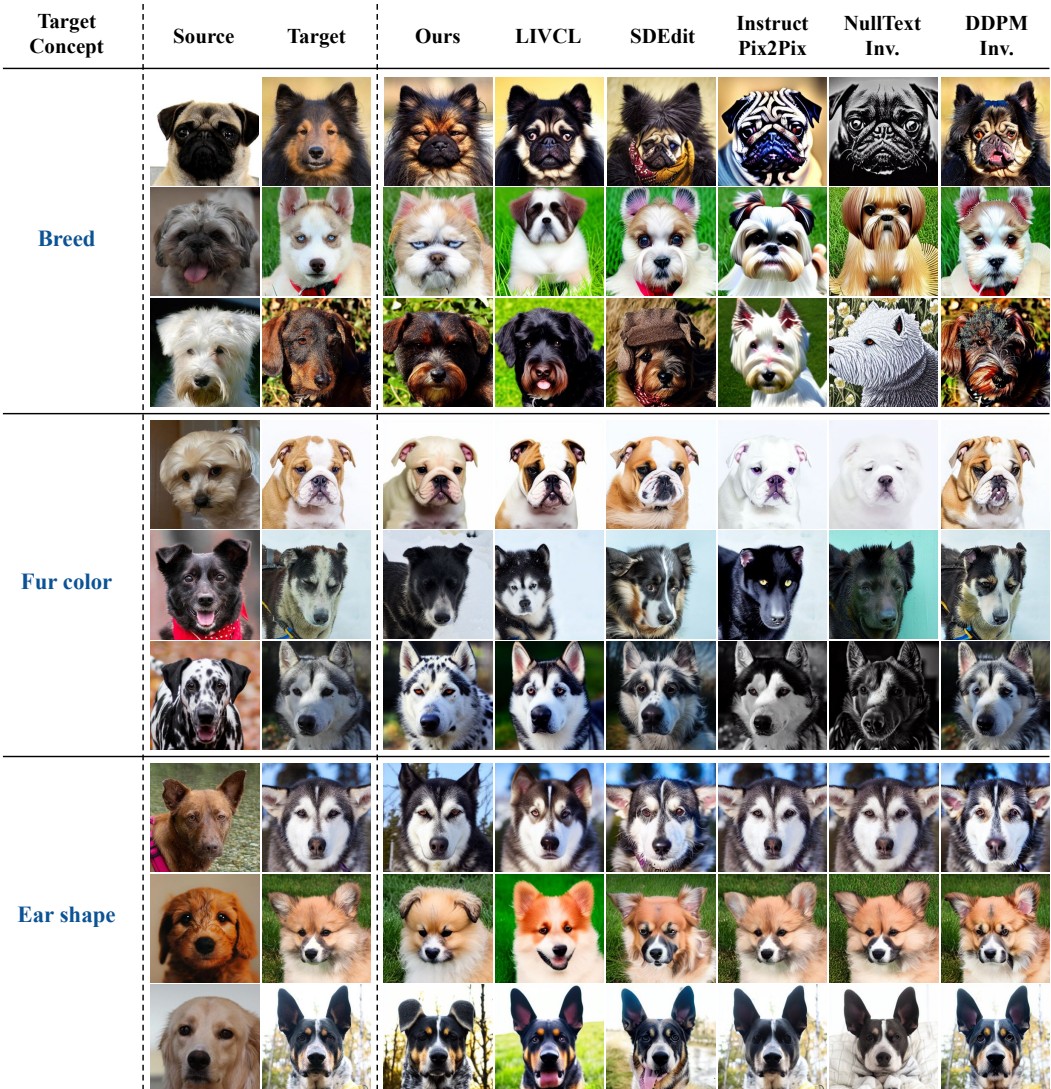

Figure 16: Additional qualitative comparison to baselines in AFHQ-Dog

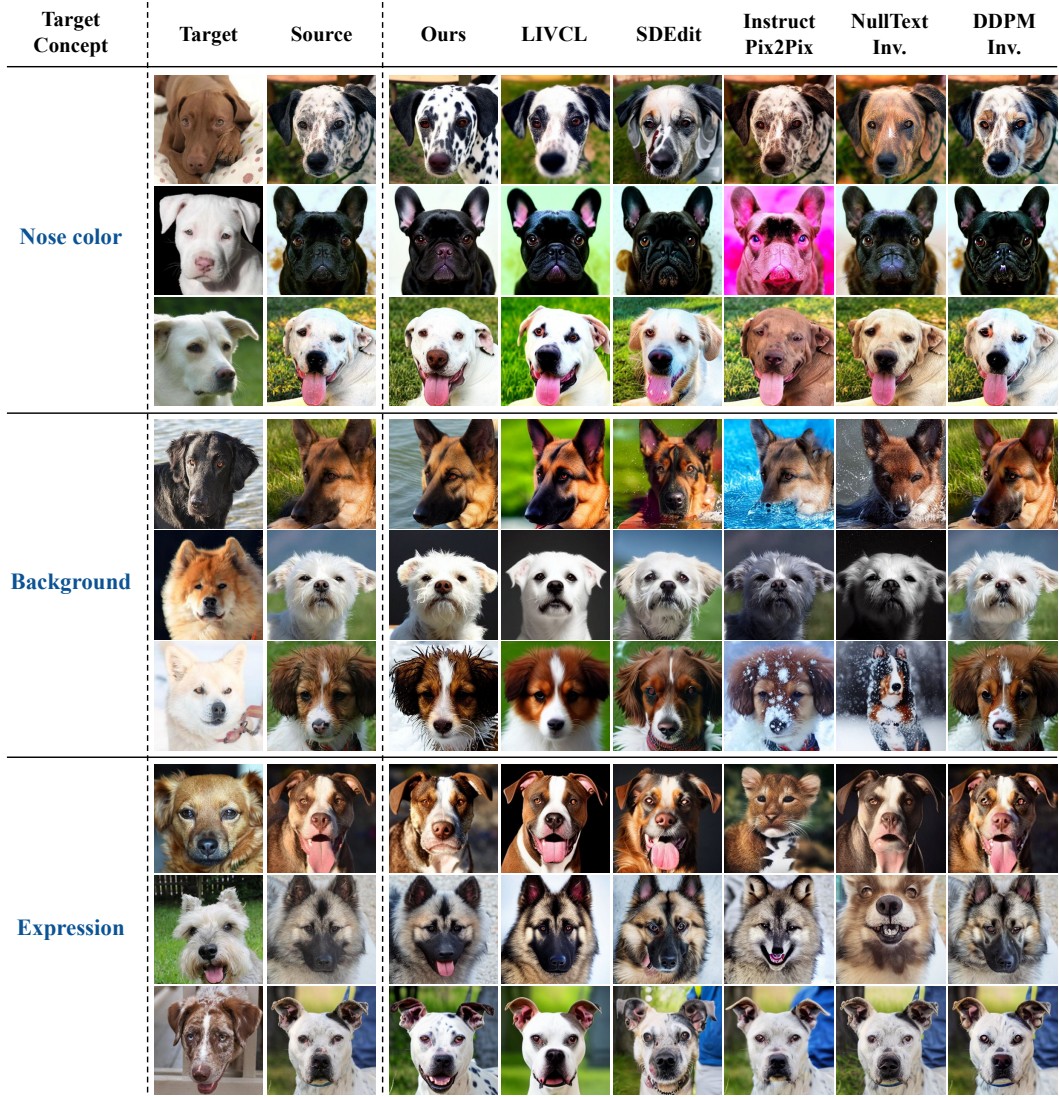

Figure 17: Additional qualitative comparison to baselines in AFHQ-Dog

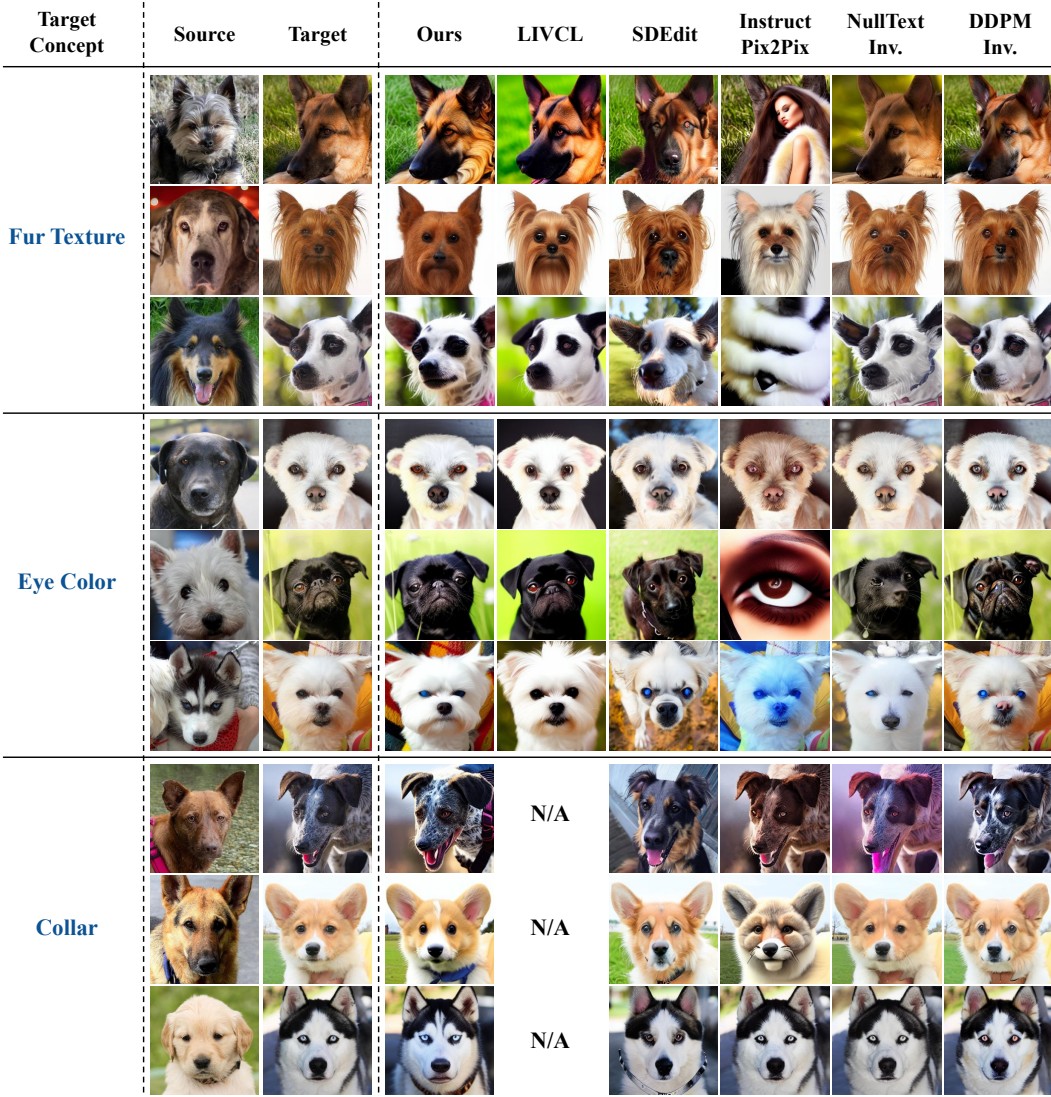

Figure 18: Additional qualitative comparison to baselines in AFHQ-Dog

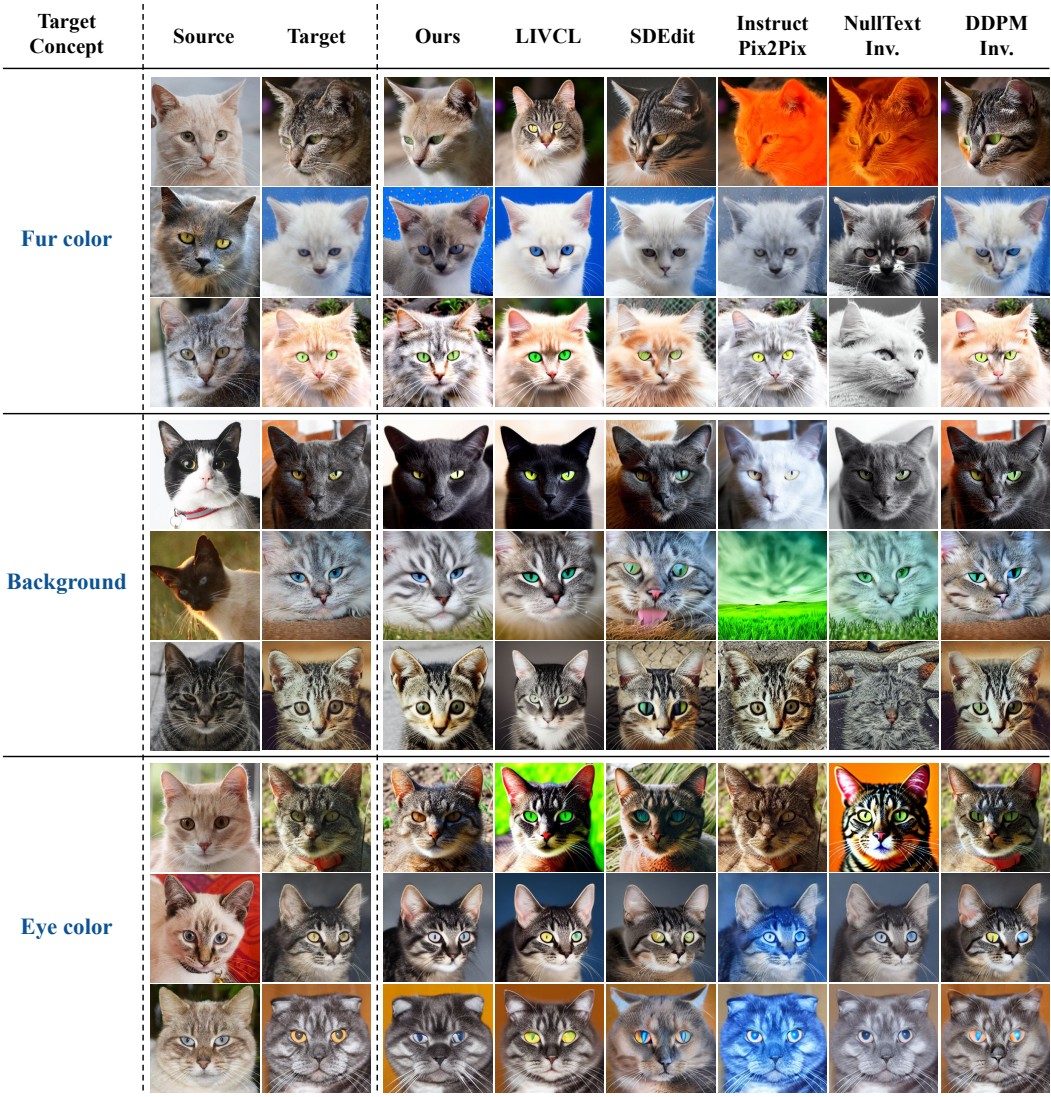

Figure 19: Additional qualitative comparison to baselines in AFHQ-Cat

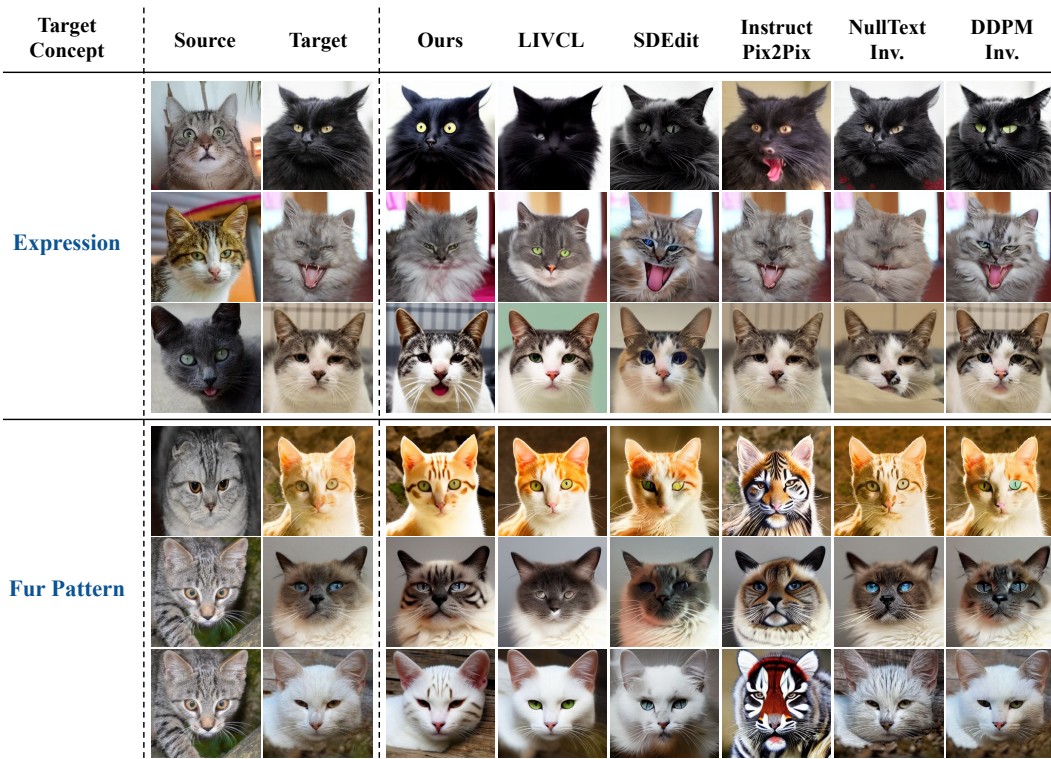

Figure 20: Additional qualitative comparison to baselines in AFHQ-Cat

### A.6.2 More Qualitative Results on Compositions from Multiple Images

We provide more qualitative results on the composition of visual concepts from multiple images in Figure 23-27. We extract $N$ distinct visual concepts from $N$ different images and replace the corresponding visual concepts of the target images with them. Our method successfully transfers multiple visual concepts to target images, which implies that each visual concept extracted from source images is disentangled along other axes.

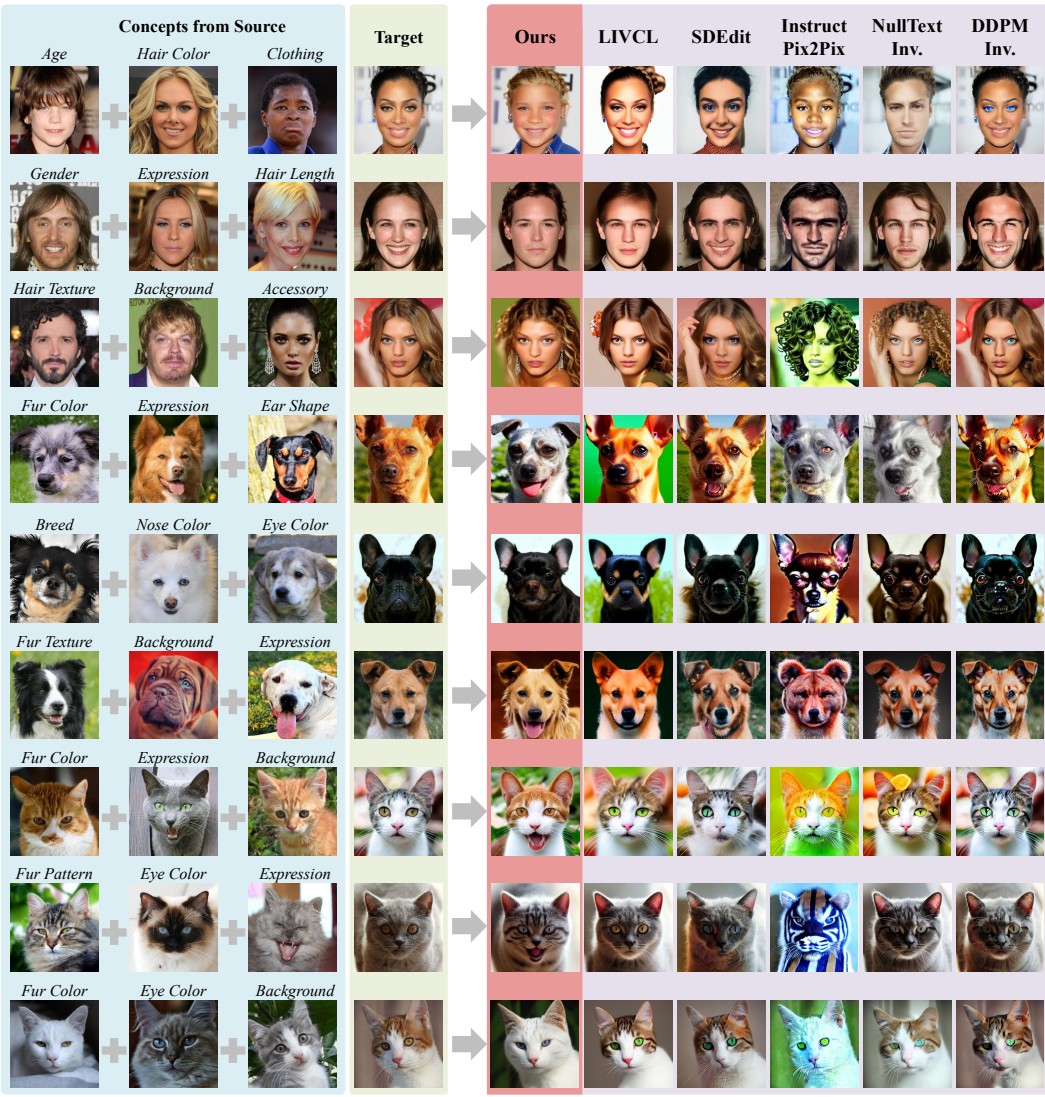

Figure 21: Compositions of visual concepts from multiple images ($N = 3$).

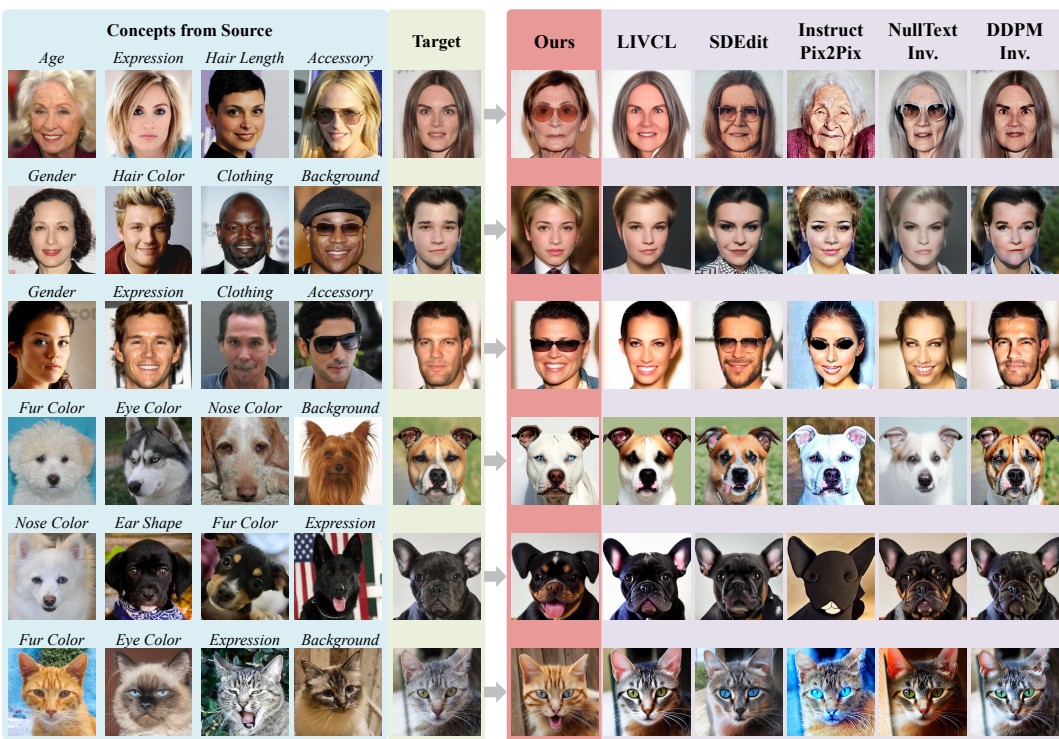

Figure 22: Compositions of visual concepts from multiple images ($N = 4$).

### A.6.3 More Qualitative Results on Visual Nuance Transfer

We provide more qualitative results on transferring visual nuance from source to target images in Figure 23-27.

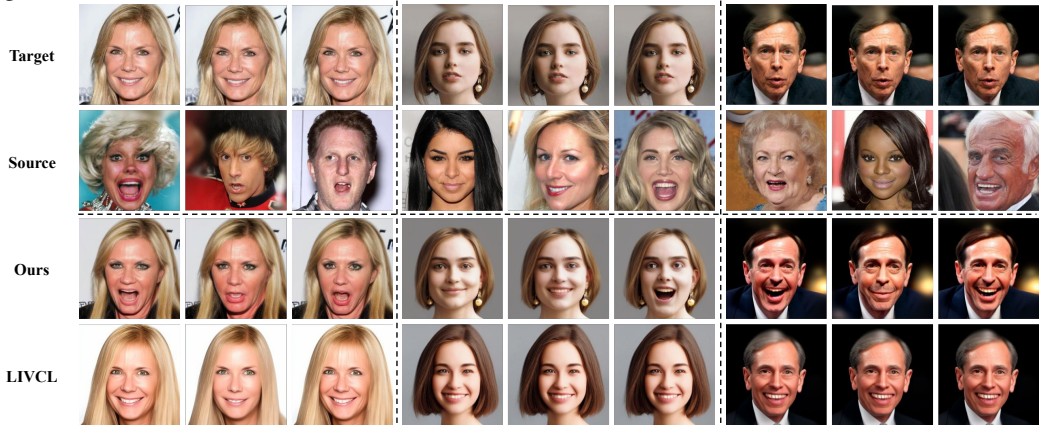

Figure 23: Transferring Visual Nuances from source to target images

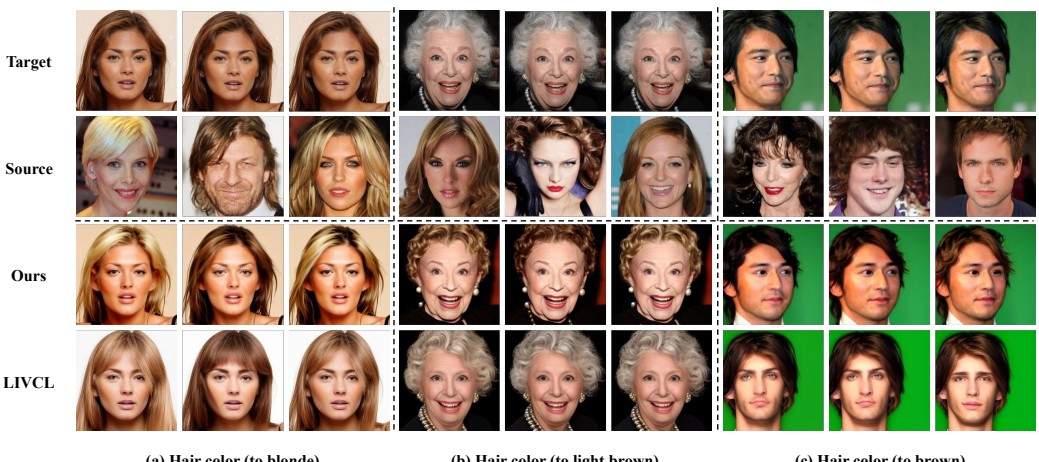

Figure 24: Transferring Visual Nuances from source to target images.

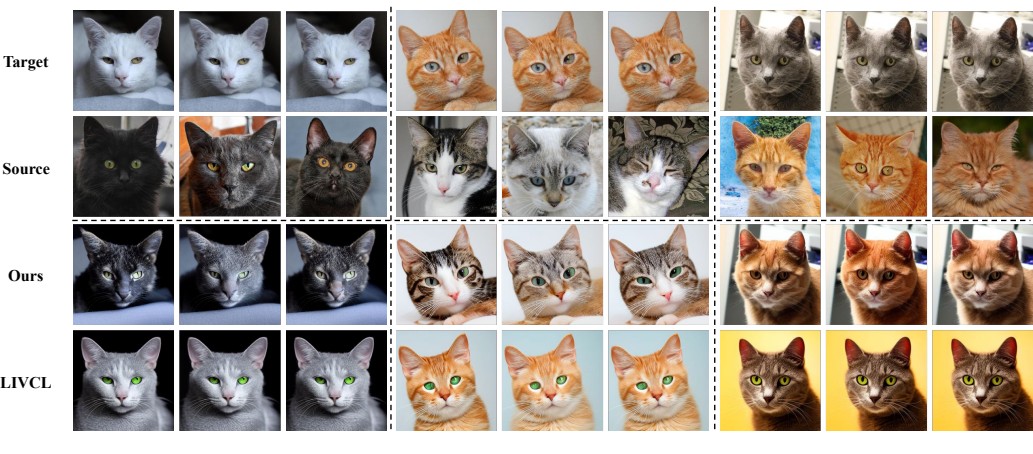

Figure 25: Transferring Visual Nuances from source to target images.

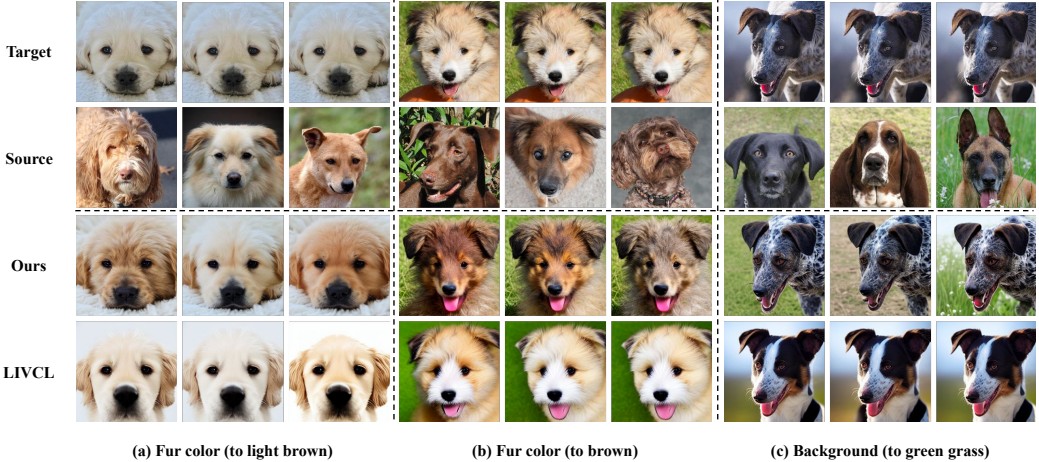

**(a) Fur color (to light brown)**  **(b) Fur color (to brown)**  **(c) Background (to green grass)**

Figure 26: Transferring Visual Nuances from source to target images.

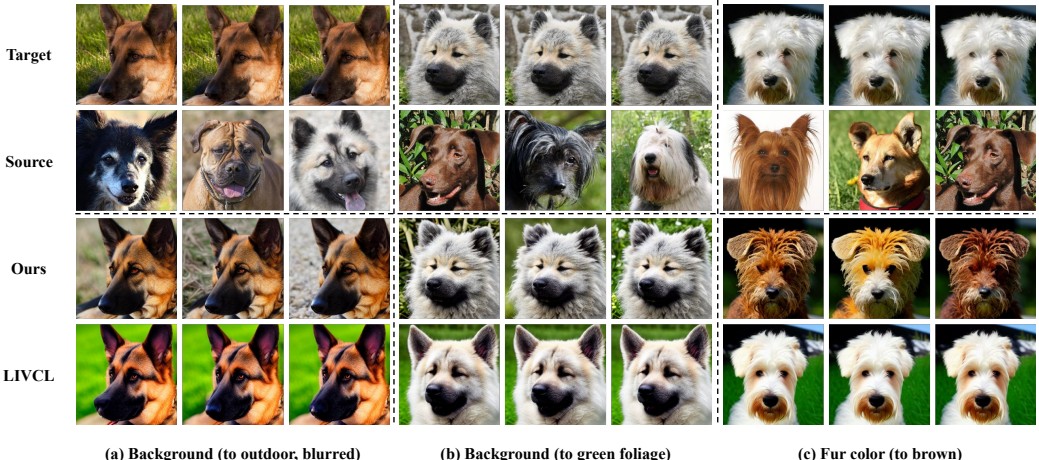

**(a) Background (to outdoor, blurred)**  **(b) Background (to green foliage)**  **(c) Fur color (to brown)**

Figure 27: Transferring Visual Nuances from source to target images.

## A.7 Computing Resources

All of our experiments are conducted on a GPU Server that consists of an Intel Xeon Gold 6230 CPU, 256GB RAM, and 8 NVIDIA RTX 6000 GPUs (with 48GB VRAM). It takes about 48 GPU hours for each dataset.

