# OpenReview forum: "Bridging the gap to real-world language-grounded visual concept learning"
_NeurIPS.cc/2025/Conference — NeurIPS 2025 poster_

### Official Review · Reviewer_5Dzc · 2025-06-19

**Clarity:** 3
**Significance:** 3
**Originality:** 2
**Rating:** 5
**Confidence:** 4

**Summary:**

This paper focuses on editing capabilities for generative models by learning a concept encoder. The authors start their method by prompting a VLM with the task of finding a set of concepts that are visible in the images. Then, they give their universal concept encoder model those concepts as input, which outputs an associated visual representation (based on DINOv2 features) that is used for conditioning a decoder (in this case, a Stable Diffusion decoder). The universal concept encoder is trained by seeking whether the generated images contain the concepts that are in the original images. The authors ran their experiments on the CelebA-HQ and AFHQ datasets and compared their methods to related works such as LIVCL, SDEdit, Instruct Pix2Pix, NullText Inv., and DDPM Inv. The evaluation is done by swapping the concepts learned between two images and observing the results. We clearly saw that the output images are more faithful to the original image while maintaining the target concept.

**Questions:**

Do you see any challenges (outside compute) for scaling your method to larger and more diverse natural images dataset? I am curious about the point you made in the appendix on identifying all the concepts being intractable. There is a limited number of words in the human languages, so this concepts are defined by mapping to English words, then this should be tractable, isn't it?

**Ethical Concerns:**

["NO or VERY MINOR ethics concerns only"]

**Final Justification:**

After reading the other reviews and the author rebuttal, I am satisfied with the discussion that has taken place. This is an interesting work that will be of interest to the community.

**Limitations:**

Addressed in the appendix.

**Paper Formatting Concerns:**

No.

**Quality:**

3

**Strengths And Weaknesses:**

Strengths:
- The paper is well written and easy to follow. The authors compare their methods with most of the related methods. The ablation study on the loss terms is also a great addition to the paper.
- The authors provide a lot of visualizations that are great to see the improvements that are led by this new method.
- Really appreciate the human study which is really important to have when comparing quality of generation across different models.

Weakness
- Lack of ablations over the prompt. Would be curious to know how the prompt influences the results.
- It would have been better to have some ablations over the VLM choice.
- The method relies on a lot of different pieces, a VLM, a vision encoder, a diffusion based decoder and the universal concept decoder that is introduced by this work. The interplay between all of those elements is not really studied in depth.
- The dataset that has been chosen for evaluation has limited diversity. For example, in the CelebA dataset there is a limited number of axis concepts. It is not clear how much this method could be scaled to the entire natural image realm.

---

> ### Author Rebuttal · Authors · 2025-07-31
>
> We appreciate the valuable comments. Below we respond to individual concerns,
>
> > Q1. Ablations over the prompt
>
> We find that the design of prompt mainly affects the granularity of the axes. For example, in CelebA-HQ dataset, simple task description without providing examples would lead to finding relatively coarse concepts such as hair and color instead of more fine-grained attributes such as hair-color, hair-length, eye color. As discovered axes are significantly different depending on the prompt, we cannot directly compare them in quantitative measures. We agree with the reviewer that investigation on prompt design might improve overall performance of our work and we leave it for future work.
>
> > Q2. Ablations on VLM choices
>
> Following the reviewer’s suggestion, we conduct two studies on the choice of VLM:
>
>
> (1) We conduct ablation study on different choices of VLMs. We consider two popular opensourced VLMs (Qwen2.5-VL and Ovfi2) that have shown high ranks in the leaderboard on reasoning benchmarks. (2) We also investigate the effect of partial observation of concept axes. While (2) is not a direct ablation study on VLMs, it can provide a controlled evaluation between the performance of VLM’s performance and representation learning quality of our framework. Since it is difficult to directly control and quantify the performance of VLMs, we instead control the quality of the output from VLMs by explicitly dropping partial axes (e.g., 10%) among the axes given by the VLMs. It is a practical scenario as it is impossible for VLM to always capture the complete axes for given scenes. To this end, we first extract concept axes per image for entire images and only 80% or 90% of axes (i.e., 10%/20% random drop) are provided to our model in the training stage. We evaluate our method on the visual concept editing task in CelebA-HQ dataset.
>
> Below Table reveals that our method is hugely robust to both the choice of VLMs and missing axes. We conjecture that this is because even though some image-related axes are missed in each example, those axes will be eventually, repeatedly exposed across the dataset. Moreover, since our compositional anchoring loss encourages the compositionality of the concept representations, our framework might be internally trained for better compositional generalization, which improves adaptation with relatively few samples. In fact, we observed that our method is capable of generating OOD samples (Figure 5 in our paper.). These results highlight the robustness of our framework regarding the performance of VLMs, and implies that our method can scale to more complex scenarios, since VLMs do not have to always capture complete axes for given scenes.
>
> === [Table 1] ===
> |  | Ours \+ InternVL2-5 | Ours \+ Qwen2.5-VL | Ours\+Ovis2 | Ours +InternVL2-5  \+ 10% drop  | Ours +InternVL2-5  \+ 20% drop  |
> |---|---|---|---|---|---|
> | CLIP Score (\uparrow) | 23.88 | 23.72 | 23.48 | 23.52 | 23.65 |
> | BLIP Score (\uparrow) | 49.58 | 48.64 | 48.35 | 48.69 | 48.61 |
>
>
> > Q3. Interplay between components
>
> We conduct ablation studies on each of the components as follows: (1) Frozen T2I decoder versus LoRA-finetuned decoder, (1) Choice of vision encoder (Dinov2 versus CLIP), (3) universal concept decoder versus shared MLP architectures.
>
>
> First, finetuning the decoder with LoRA does not affect the overall performance. Large scale pretrained T2I models are known to be already learned expressive data priors on natural images, which helps faster training of the generation model. Thus the frozen T2I model does not bottleneck our framework. Replacing Dinov2 encoder with CLIP encoder leads to significant performance drop. This is because CLIP is only trained for text-alignment and thereby it’s discriminative property of representation is likely to be inferior to recent self-supervised learning methods (Dinov2).
>
>
> Lastly, we replace our universal concept encoder with shared MLP architecture, which is naive version of axis-agnostic encoder. Specifically, visual feature is first represented as vector through mean pooling, and it is concatenated with axis embeddings followed by MLPs to encode concept representations. To make this encoder generally work for diverse concept axes, we shared this MLP layer for all of the axes. However, this modification causes severe performance drop. We conjecture that this is because shared MLP encodes each concept independently, possibly blocking complex interaction between visual concepts. It clearly highlights the effectiveness of our universal concept encoders.
>
>
> === [Table 2] ===
>
> |  | CLIP | BLIP |
> |---|---|---|
> | Frozen T2I  + Dinov2 + UCE (Ours) | 23.88 | 49.58 |
> | LoRA-finetuned T2I + Dinov2 + UCE | 23.67 | 49.62 |
> | Frozen T2I + CLIP + UCE | 22.03 | 46.21 |
> | Frozen T2I  + Dinov2 + Shared MLP | 21.63 | 46.86 |
>
>
>
> > Q4. Experiments on diverse dataset
>
> To validate our framework on larger, diverse, and unstructured data, we randomly sampled 20 ImageNet classes, spanning animals (e.g., tree frog, American black bear, sulphur butterfly, giant panda), everyday objects (e.g., padlock, grand piano, motor scooter), scenes (e.g., boathouse, water tower), and etc,  yielding around 28k training images (~1.4k images per class). This is a significantly more challenging scenario as it contains diverse image-specific visual concepts and unstructured images. Across this dataset, our method successfully discovers both coarse and fine-grained across different categories. Spanning from primitive concepts like color, material, shape, it also discovers fine-grained axes such as wing color/shape, body color of butterflies and sail color, boat color, sky condition for boat images. It also captures categorical attributes, such as species, vehicle type, fruit type. Although our dataset does not include human-centered images as in CelebA-HQ dataset, our framework still robustly captures human-specific concepts such as age, gender, hair color/length.
> For quantitative assessment, we conduct visual concept editing on the 50 most frequent axes (please refer to response to Q2. of Reviewer 4ztc  for top-50 frequent axes) and report CLIP/BLIP Scores against text-conditioned editing baselines. Manually predefined-axis based approach (LICVL [1]) becomes infeasible here, since a fixed set of axes fails to cover a wide range of concept space. Even within the same category, diverse image-specific concepts would appear, for example, humans, animals and plants may appear in motor_scooter class images, making it difficult to predefine a fixed set of concept axes representing the entire images. It highlights the effectiveness of our method, as our framework can adaptively decompose the given scene into image-related axes specified by VLMs.
> Below Table demonstrates superior performance of our method compared to text-conditioned editing baselines. As the discovered visual concepts become more detailed such as body color of butterflies or stem color of flowers, most of the text-conditioned editing baselines fail to perform faithful editing. Also, we find that our framework is capable of generating OOD samples such as green bears/pandas or a scooter floating above the sea. We will add qualitative results in our main paper.
>
>
> |  | Pix2Pix | NullText Inversion | SDEdit | DDPM Inversion | Ours |
> |---|:---:|:---:|:---:|:---:|:---:|
> | CLIP Score ($\uparrow$) | 19.83 | 18.93 | 19.54 | 24.32. | **25.10** |
> | BLIP Score ($\uparrow$) | 38.34 | 34.10 | 38.10 | 46.70 | **47.44** |
>
>
> > Q5. Challenges (outside compute) for scaling your method to larger and more diverse natural images dataset
>
> Yes, mapping visual concepts to verbal words is an important motivation of our work. As there could be infinitely many concept axes in real-worlds, aligning with our perception and those semantic representations are crucial and effective way to describe the complex world with a feasible number of concepts. We found that rare and too fine-grained concepts such as the exact brand name of the calculator are hard to be trained, as our encoder rarely meets those concepts and hardly get signals from compositional anchoring loss.
>
>
> **Reference**
>
> [1] Lee et al., language-informed visual concept learning, in ICLR 24.

---

> > ### Comment · Reviewer_5Dzc · 2025-08-05
> >
> > Thank you for answering my concerns. I really appreciate the additional experiments over the Imagenet subset. The qualitative OOD results should definitively be added in the paper. It's not really surprising than CLIP perform indeed worse than DinoV2 as vision encoder but it's great to have the empirical validation. Thank you also for doing the ablation over the VLM choice as well as on dropping axes.
> >
> > Overall, this is a good work, I am increasing my score to 5.

---

> ### Author Response · Authors · 2025-08-06
> **Reply to Reviewer 5Dzc**
>
> We sincerely appreciate the reviewer's valuable comments. Thanks to the reviewer's thoughtful comments and feedback, additional experiments on ablation studies and diverse dataset make our work more comprehensive. We will add qualitative OOD samples and additional experimental results in our revised paper.

---

### Official Review · Reviewer_xMmD · 2025-07-01

**Clarity:** 3
**Significance:** 2
**Originality:** 3
**Rating:** 5
**Confidence:** 3

**Summary:**

This paper proposes a unified framework for compositional visual concept learning from real-world images along user-defined semantic axes. A universal concept encoder is introduced to extract axis-wise latent representations from images, guided by text embeddings. These representations are used to condition a frozen text-to-image (T2I) diffusion model, enabling compositional image editing and generation. The method is trained using a combination of diffusion reconstruction loss, compositional consistency loss (via representation swapping), and text regression loss.

**Questions:**

Please refer to Weaknesses.

**Ethical Concerns:**

["NO or VERY MINOR ethics concerns only"]

**Final Justification:**

As my concern has been mostly resolved, I have decided to raise my score.

**Limitations:**

Please refer to Weaknesses.

**Quality:**

3

**Strengths And Weaknesses:**

### **Strengths**

1. **Clear motivation and task definition**: The authors clearly articulate the goal of axis-guided concept decomposition and recombination, addressing the need for interpretable and composable visual representations.

2. **Model modularity and flexibility**: The universal concept encoder is compatible with arbitrary concept axes defined in natural language, allowing potential extensibility.

3. **Loss design for disentanglement**: The proposed compositional consistency loss is novel and aligns well with the goal of achieving axis-specific representations.

4. **Qualitative and visual interpretability**: The model demonstrates controllable image generation through partial axis manipulation, which is intuitive and informative.


### Weaknesses and Concerns

**1.Dependence on a single VLM for axis generation**:
The semantic axes in this framework are derived using a single vision-language model. It inherently limits the diversity and robustness of the generated axes. The process also depends on the quality and coverage of the VLM’s representations. Incorporating multiple VLMs or leveraging fixed reference taxonomies (e.g., attribute banks or external ontologies) could improve the generality and semantic coverage of the axis discovery process.

**2.Assumption of axis independence**: The model enforces disentanglement by treating each axis as independent, without modeling correlations or dependencies between semantic axes, which are common in natural images.

**3.Evaluation scope is narrow**: The paper primarily evaluates on image editing and attribute recomposition tasks. It does not compare against related methods in compositionality, zero-shot learning, or attribute-conditioned generation, making it difficult to assess general effectiveness.

**4.Frozen T2I model as a bottleneck**: The use of a fixed text-to-image diffusion model constrains the upper bound of generation quality and may obscure the actual representational quality of the learned embeddings.

---

> ### Author Rebuttal · Authors · 2025-07-31
>
> We appreciate the valuable comments. Below we respond to individual concerns,
>
> > Q1. Dependence on a single VLM for axis generation.
>
> We appreciate the valuable feedback and suggestions. We agree with the reviewer that the performance of VLM would largely affect our framework. While incorporating multiple VLMs or employing fixed reference taxonomies would be an interesting and effective idea, we found that it is not trivial for extend and formulate the method during the rebuttal period. Thereby we leave it for our important future work.
>
> Instead, we provide a clarification that our method is quite robust to the performance of VLMs. We perform two additional experiments: We measure performance on (1) different choices of VLMs, and (2) randomly drop partial axes during training. For the first experiment, we consider two additional opensourced VLMs (Qwen2.5-VL and OVis2) that have shown high ranks in the leaderboard on reasoning benchmarks. For the second experiment, as it is difficult to control and quantify how much of the axes are missed by VLM, we instead investigate the quality of axes provided by VLMs. We explicitly drop partial axes (10% and 20%) from the axes provided by VLMs to simulate imperfect prediction of VLMs. To this end, we first extract concept axes per image for entire images and only 80% or 90% of axes (i.e., 10%/20% random drop) are provided to our model in the training stage. We evaluate our method on the visual concept editing task in CelebA-HQ dataset.
>
>
> The below Table reveals that our method is hugely robust to both the choice of VLMs and missing axes. We conjecture that this is because even though some image-related axes are missed in each example, those axes will be eventually, repeatedly exposed across the dataset. Moreover, since our compositional anchoring loss encourages the compositionality of the concept representations, our framework might be internally trained for better compositional generalization. In fact, we observed that our method is capable of generating OOD samples (Figure 5 in our paper.). These results highlight the robustness of our framework regarding the performance of VLMs, and implies that our method can scale to more complex scenarios, since VLMs do not have to always capture complete axes for given scenes.
>
>
> |  | Ours \+ InternVL2-5 | Ours \+ Qwen2.5-VL | Ours\+Ovis2 | Ours +InternVL2-5  \+ 10% drop  | Ours +InternVL2-5  \+ 20% drop  |
> |---|---|---|---|---|---|
> | CLIP Score (\uparrow) | 23.88 | 23.72 | 23.48 | 23.52 | 23.65 |
> | BLIP Score (\uparrow) | 49.58 | 48.64 | 48.35 | 48.69 | 48.61 |
>
>
>
>
> > Q2. Assumption of axis independence
>
>  We would like to clarify that we do not impose statistical independence between the axes.
> The only constraint imposed by our framework is compositionality of visual concept axes. Therefore, in our experimental result, our method often disentangle highly correlated factors such as breed of the dog and folding of the ear shape and generates OOD samples (Figure 5 of our main paper). However, we agree with the reviewer that handling the correlation between semantic axes is indeed an important problem and leave it to important direction for future work.
>
> > Q3. Evaluation scope is narrow
>
>  We believe our evaluation protocol on image editing is already implicitly measuring the compositionality of the learned representations. As we compose two or more sets of concept representations and measure alignment between resulting composite and those composite attributes, it can be interpreted as measuring the compositionality of the representation. We appreciate the valuable comment from the reviewer and will find the proper evaluation protocol for measuring compositionality and update it in our main paper.
>
>
> > Q4. Frozen T2I model as a bottleneck.
>
> We agree that fine-tuning the T2I backbone, e.g. via LoRA, would boost overall image fidelity, but we would like to emphasize that even with a fully frozen diffusion model, our learned embeddings already achieve high-quality, semantically accurate edits. This implies that our framework does not require to train huge generative models every time or store task-specific parameters (e.g., LoRA) for T2I model, which eases the training process of our framework. To support our claim, we finetune our T2I model with LoRA, and report CLIP/BLIP Score for visual concept editing tasks. We find that the effect of finetuning LoRA seems to be negligible in terms of concept learning.
>
>
> Nevertheless, we agree with the reviewer that joint training of the T2I model would have clear benefits in image generation or faster adaptation to unseen samples and tasks. We appreciate the valuable comment and leave it for the future work.
>
>
> === [Table 1] ===
>
> |  | CLIP | BLIP |
> |---|---|---|
> | ours + frozen T2I | **23.88** | 49.58 |
> | ours + LoRA-tuned T2I | 23.67 | **49.62** |

---

> > ### Comment · Reviewer_xMmD · 2025-08-07
> >
> > As my concern has been mostly resolved, I have decided to raise my score.

---

> > > ### Author Response · Authors · 2025-08-09
> > > **Reply to Reviewer xMmD**
> > >
> > > We are glad to hear that our response has addressed the reviewer’s concerns and sincerely appreciate the thoughtful comments again. In the revised paper, we will include the results and discussion from the additional experiments on the robustness analysis of the VLM and the fine-tuned T2I model.

---

> ### Author Response · Authors · 2025-08-06
>
> Dear Reviewer xMmD,
>
> We would like to sincerely thank the reviewer again for their valuable comments. We kindly follow up to inquire whether our response has sufficiently addressed the reviewer’s questions. Should the reviewer have any further queries or concerns, we would be happy to discuss.

---

### Official Review · Reviewer_4ztc · 2025-07-02

**Clarity:** 2
**Significance:** 1
**Originality:** 3
**Rating:** 4
**Confidence:** 4

**Summary:**

The authors propose a scalable framework that adaptively identifies image-related concept axes and grounds visual concepts along these axes in real-world scenes. Specifically, they propose to adaptively identify image-related axes with a pretrained VLM, and design a universal concept encoder that adaptively binds visual features to these axes.

**Questions:**

According to the weaknesses I previously highlighted, addressing the following points could potentially change my opinion:
1. Provide a thorough analysis and discussion regarding the reliance on VLMs.
2. Try to modify the current architecture or clarify that it is not merely a building-block work.
3. Conduct comprehensive experiments on more complex datasets and compare the results with SOTA approaches.

**Ethical Concerns:**

["NO or VERY MINOR ethics concerns only"]

**Final Justification:**

According to the responses, the authors have addressed my primary concerns regarding the necessity of applying the framework to more complex scenes and datasets. Even I maintain that the significant reliance on VLM may impose limitations on the model's upper bound, I am inclined to raise my score to 4, reflecting my appreciation for the authors' current stage of work. I look forward to the inclusion of additional visual results and a more in-depth discussion of VLM's perceptual capabilities, as promised by the authors.

**Limitations:**

Yes

**Quality:**

2

**Strengths And Weaknesses:**

**Strengths**

1. The authors propose visual concept learning from the perspective of disentangled representation learning is a sensible approach.

2. The results indicate that the model possesses a certain degree of attribute editing and disentanglement capability.

**Weaknesses**

1. The heavy reliance on Visual Language Models (VLM) diminishes the contribution of this work. Prompting VLM to define attribute axes and values is straightforward, and it limits the model to the upper-bound of the VLM. This approach is based on the assumption that VLMs or MLLMs can accurately perceive the attribute axes for all scenes. While pre-trained VLMs may perform well with simple objects, they may struggle in more complex scenes.

2. Following the first weakness, I expect the paper to provide empirical evidence in more complex scenes; however, it appears that only visual results from facial datasets (CelebA and AFHQ) are presented. The authors should include additional results from challenging scenes, such as cars, buildings, and animals, to demonstrate the generalizability of the model.

3. As a concept/representation learning approach, the combination of VLM prompting, Q-former encoder, and Compositional Consistency Loss is too shallow and simplistic. I believe the authors should delve deeper into how VLM perceives attributes rather than merely using it as a tool. Additionally, I expect a more specially designed autoencoder architecture for concept learning to meet the standards of top-tier venues.

---

> ### Author Rebuttal · Authors · 2025-07-31
>
> We appreciate the valuable comments. Below we respond to individual concerns,
>
> > Q1. Reliance on Visual Language Models (VLM)
>
> We agree with the reviewer that VLM cannot always perceive a complete set of concept axes. Nevertheless, we believe that the use of VLM provides a significant advantage in complex scenarios, where it is almost infeasible to manually annotate diverse image-specific concepts. Although VLM would not be perfect, it can still capture meaningful and diverse visual concepts ( Please refer to our response to Q2 below.)
>
> In addition, we conduct additional experiments to investigate the robustness of our framework to VLMs. We perform two additional experiments: We measure performance on (1) different choices of VLMs, and (2) randomly drop partial axes from total axes provided by the VLMs. For the first experiment, we consider two additional opensourced VLMs (Qwen2.5-VL and OVis2) that have shown high ranks in the leaderboard on reasoning benchmarks. For the second experiment, as it is difficult to control and quantify how much of the axes are missed by VLM, we instead investigate the performance differences when we explicitly drop partial axes (10% and 20%) from the axes provided by VLMs. To this end, we first extract concept axes per image for entire images and only 80% or 90% of axes (i.e., 10% and 20% of random drop) are provided to our model in the training stage. We evaluate our method on the visual concept editing task in CelebA-HQ dataset.
>
>
> The below Table reveals that our method is hugely robust to both the choice of VLMs and missing axes. We conjecture that this is because even though some image-related axes are missed in each example, those axes will be eventually, repeatedly exposed across the dataset. Moreover, since our compositional anchoring loss encourages the compositionality of the concept representations, our framework might be internally trained for better compositional generalization. In fact, we observed that our method is capable of generating OOD samples (Figure 5 in our paper.). These results highlight the robustness of our framework regarding the performance of VLMs, and implies that our method can scale to more complex scenarios, since VLMs do not have to always capture complete axes for given scenes.
>
> === [Table 1] ===
>
> |  | Ours \+ InternVL2-5 | Ours \+ Qwen2.5-VL | Ours\+Ovis2 | Ours +InternVL2-5  \+ 10% drop  | Ours +InternVL2-5  \+ 20% drop  |
> |---|---|---|---|---|---|
> | CLIP Score ($\uparrow$) | 23.88 | 23.72 | 23.48 | 23.52 | 23.65 |
> | BLIP Score ($\uparrow$) | 49.58 | 48.64 | 48.35 | 48.69 | 48.61 |
>
> > Q2. Experiments on more complex dataset.
>
> Following the suggestion of the reviewer, we validate our framework on larger, diverse, and unstructured data. We randomly sampled 20 ImageNet classes, spanning animals (e.g., tree frog, American black bear, sulphur butterfly, giant panda), everyday objects (e.g., padlock, grand piano, motor scooter), and scenes (e.g., boathouse, water tower) yielding around 28k training images (~1.4k images per class). This is a significantly more challenging scenario as it contains diverse image-specific visual concepts and unstructured images.
>
> Across this dataset, our method successfully discovers both coarse and fine-grained across different categories. Spanning from primitive concepts like color, material, shape, it also discovers fine-grained axes such as wing color/shape, body color of butterflies and sail color, boat color, sky condition for boat images. It also captures categorical attributes, such as species, vehicle type, fruit type. Although our dataset does not include human-centered images as in CelebA-HQ dataset, our framework still robustly captures human-specific concepts such as age, gender, hair color/length.
>
> For quantitative assessment, we conduct visual concept editing on the 50 most frequent axes (We provide top-50 frequent axes at the end of the response) and report CLIP/BLIP Scores against text-conditioned editing baselines. Manually predefined-axis based approach (LICVL [1]) becomes infeasible here, since a fixed set of axes fails to cover a wide range of concept space. Even within the same category, diverse image-specific concepts would appear, for example, humans, animals and plants may appear in motor_scooter class images, making it difficult to predefine a fixed set of concept axes representing the entire images. It highlights the effectiveness of our method, as our framework can adaptively decompose the given scene into image-related axes specified by VLMs.
>
> Below Table demonstrates superior performance of our method compared to text-conditioned editing baselines. As the discovered visual concepts become more detailed such as body color of butterflies or stem color of flowers, most of the text-conditioned editing baselines fail to perform faithful editing. Also, we find that our framework is capable of generating OOD samples such as green bears/pandas or a scooter floating above the sea. We will add qualitative results in our main paper.
>
>
> |  | Pix2Pix | NullText Inversion | SDEdit | DDPM Inversion | Ours |
> |---|:---:|:---:|:---:|:---:|:---:|
> | CLIP Score ($\uparrow$) | 19.83 | 18.93 | 19.54 | 24.32. | **25.10** |
> | BLIP Score ($\uparrow$) | 38.34 | 34.10 | 38.10 | 46.70 | **47.44** |
>
> > Q3. Try to modify the current architecture or clarify that it is not merely a building-block work.
>
> We would like to first clarify that our main scope is not designing a specific model architecture to improve the performance. Instead, our main contribution is to build a framework that can adaptively capture image-specific visual concepts in real-world scenes.
> To this end, we focus on designing our framework to address such challenges, while leaving other components general for further applicability.
>
>
> Additionally, to justify the choice of each component, we conduct ablation studies as follows: (1) Frozen T2I decoder versus LoRA-finetuned decoder, (1) Choice of vision encoder (Dinov2 versus CLIP), (3) universal concept decoder versus shared MLP architectures.
>
>
> First, finetuning the decoder with LoRA does not affect the overall performance. Large scale pretrained T2I models are known to be already learned expressive data priors on natural images, which helps faster training of the generation model. Thus the frozen T2I model does not bottleneck our framework. Replacing Dinov2 encoder with CLIP encoder leads to significant performance drop. This is because CLIP is only trained for text-alignment and thereby it’s discriminative property of representation is likely to be inferior to recent self-supervised learning methods (Dinov2).
>
>
> Lastly, we replace our universal concept encoder with shared MLP architecture, which is naive version of axis-agnostic encoder. Specifically, the visual feature is first represented as vector through mean pooling, and it is concatenated with axis embeddings followed by MLPs to encode concept representations. To make this encoder generally work for diverse concept axes, we shared this MLP layer for all of the axes. However, this modification causes severe performance drop. We conjecture that this is because shared MLP encodes each concept independently, possibly blocking complex interaction between visual concepts. It clearly highlights the effectiveness of our universal concept encoders.
>
> === [Table 2] ===
> |  | CLIP | BLIP |
> |---|---|---|
> | Frozen T2I  + Dinov2 + UCE (Ours) | 23.88 | 49.58 |
> | LoRA-finetuned T2I + Dinov2 + UCE | 23.67 | 49.62 |
> | Frozen T2I + CLIP + UCE | 22.03 | 46.21 |
> | Frozen T2I  + Dinov2 + Shared MLP | 21.63 | 46.86 |
>
>
>
>
> **Reference**
>
> [1] language-informed visual concept learning, in ICLR 24.
>
> **Top-50 axes discovered by our framework** :
>
> [subject_type, background, color, material, shape, species, size, type, vehicle_type, environment, position, cap_color, gender, age, object_type, hair_color, hair_length, lighting, accessory, location, water_body, expression, clothing, activity, sky_condition, eye_color, body_color, windows, wing_shape, fur_texture, stem_color, surroundings, sail_color, seat_color, wing_color, wing_pattern, fruit_type, color_pattern, surface_texture, limbs, flower_color, style, design, wall_color, texture, brand, container_count, flooring, cap_texture, boat_color]

---

> > ### Comment · Reviewer_4ztc · 2025-08-04
> >
> > Thanks for the response. The authors have addressed my primary concerns regarding the necessity of applying the framework to more complex scenes and datasets. Even I maintain that the significant reliance on VLM may impose limitations on the model's upper bound, I am inclined to raise my score to 4, reflecting my appreciation for the authors' current stage of work. I look forward to the inclusion of additional visual results and a more in-depth discussion of VLM's perceptual capabilities, as promised by the authors.

---

> ### Author Response · Authors · 2025-08-06
> **Reply to Reviewer 4ztc**
>
> We sincerely appreciate the reviewer's detailed feedback and thoughtful comments. Thanks to the reviewer's valuable suggestions, our work has became more comprehensive. As promised, we will include additional visual results and provide a more in-depth discussion of the VLM's perceptual capabilities in our revised paper.

---

### Official Review · Reviewer_vNjW · 2025-07-03

**Clarity:** 3
**Significance:** 2
**Originality:** 2
**Rating:** 3
**Confidence:** 2

**Summary:**

The paper presents an approach for visual concept learning that automatically discovers an open-ended set of semantically meaningful axes along with images in a dataset can vary.  It leverages VLMs to both propose the axes of variation and ground these concepts.   The discovered axes permit more flexible and precise image editing, as well as compositional generalization of multiple image analogies at the same time.

**Questions:**

Any further information the authors could give that addresses the weaknesses above would be appreciated!

**Ethical Concerns:**

["NO or VERY MINOR ethics concerns only"]

**Final Justification:**

I appreciate the extra simulations and results but I don't feel they strongly show a major advantage over other work. They also don't negatively impact my impression. So I will keep my score the same.

**Limitations:**

More details on limitations, linked up to analyses of specific kinds of errors that the system makes, would be helpful.

**Quality:**

2

**Strengths And Weaknesses:**

Strengths: The paper's motivation is clear and reasonable.  The results show both qualitative and quantitative improvements over baselines. I am not an expert in the paper's area and so I can't really judge the novelty or level of contribution.

Weaknesses: The paper doesn't offer a fundamentally new approach or a fundamental, principles-driven or theory driven approach.  It's all about prompting and tuning the VLM.  It's hard to see what I learn from the work of general or broader interest.

The evaluations are based on relatively limited datasets, drawn from very limited domains (faces and animals). On both datasets, there are already well-defined axes during data collection (and VLMs/LLMs have seen those examples). It’s unclear how the approach could generalize to more unstructured datasets.

Also, on both faces and animals, images can be aligned to a common 2D reference frame which might make concept discovery and independent editing easier – and unrepresentative of what we would expect to see in a larger and more diverse dataset.

Another potential limitation is that the method relies on VLMs proposing the semantic meaningful axes and attributes for individual images (starting with asking VLMs to categorize all attributes in the input images). There seems to no robustness analysis: what will happen if the VLM misses an axis, or misses an attribute, or how often this happens or under what circumstances.

---

> ### Author Rebuttal · Authors · 2025-07-31
>
> We appreciate the valuable comments. Below we respond to individual concerns,
>
>
> >Q1. The paper doesn't offer a fundamentally new approach or a fundamental, principles-driven or theory driven approach.
>
> We would like to first clarify the contribution of our work. It is generally difficult for machine learning algorithms to capture underlying visual concepts in real-world scenarios, since each image consists of very diverse, image-specific axes. While it is infeasible to disentangle all of those factors without supervision, a recent breakthrough has been made by incorporating a vision language model to ground each concept axis with textual description [1]. However, this method still requires manual definition of concept axes and thereby limits its application to complex real-world scenes having infinitely many axes.
>
> In this work, we address such challenges by designing our framework to adaptively perceive image-related axes aided by VLMs. We believe the problem setting and overall formulation itself is an important contribution to the field as it provides a first step towards visual concept learning in a general, real-world scenes. Moreover, we also propose novel objective (compositional anchoring loss) to disentangle each representation along image-specific axes, and propose a simple yet effective architecture (Universal Concept Encoder) to encode a varying number of concept representations. Our ablation study (Table 3 in our main paper) supports that novel objectives is essential component of our framework. Moreover, we provide additional experiments on more diverse and complex natural image datasets, which highlights the effectiveness and scalability of our framework (Please see our response to Q2.)
>
>
> >Q2. Experiments on more diverse and unstructured datasets
>
> To validate our framework on larger, diverse, and unstructured data,
> we randomly sampled 20 ImageNet classes, spanning animals (e.g., tree frog, American black bear, sulphur butterfly, giant panda), everyday objects (e.g., padlock, grand piano, motor scooter), and scenes (e.g., boathouse, water tower) yielding around 28k training images (~1.4k images per class). This is a significantly more challenging scenario as it contains diverse image-specific visual concepts and unstructured images.
>
>
> Across this dataset, our method successfully discovers both coarse and fine-grained across different categories. Spanning from primitive concepts like color, material, shape, it also discovers fine-grained axes such as wing color/shape, body color of butterflies and sail color, boat color, sky condition for boat images. It also captures categorical attributes, such as species, vehicle type, fruit type. Although our dataset does not include human-centered images as in CelebA-HQ dataset, our framework still robustly captures human-specific concepts such as age, gender, hair color/length.
>
> For quantitative assessment, we conduct visual concept editing on the 50 most frequent axes (We provide top-50 frequent axes at the end of the response) and report CLIP/BLIP Scores against text-conditioned editing baselines. Manually predefined-axis based approach (LICVL [1]) becomes infeasible here, since a fixed set of axes fails to cover a wide range of concept space. Even within the same category, diverse image-specific concepts would appear, for example, humans, animals and plants may appear in motor_scooter class images, making it difficult to predefine a fixed set of concept axes representing the entire images. It highlights the effectiveness of our method, as our framework can adaptively decompose the given scene into image-related axes specified by VLMs.
>
> Below Table demonstrates superior performance of our method over text-conditioned editing baselines. As the discovered visual concepts become more detailed such as body color of butterflies or stem color of flowers, most of the text-conditioned editing baselines fail to perform faithful editing. Also, we find that our framework is capable of generating OOD samples such as green bears/pandas or a scooter floating above the sea. We will add extensive qualitative results in our main paper.
>
> === [Table 1] ===
> |  | Pix2Pix | NullText Inversion | SDEdit | DDPM Inversion | Ours |
> |---|:---:|:---:|:---:|:---:|:---:|
> | CLIP Score ($\uparrow$) | 19.83 | 18.93 | 19.54 | 24.32. | **25.10** |
> | BLIP Score ($\uparrow$) | 38.34 | 34.10 | 38.10 | 46.70 | **47.44** |
>
>
> > Q3. Robustness analysis on on VLMs
>
>
>
>
>
>  To analyze the robustness of our framework to VLM, we conduct two additional experiments: We measure performance on (1) different choices of VLMs, and (2) randomly drop partial axes from total axes provided by the VLMs.
>
>
> For the first experiment, we consider two additional opensourced VLMs (Qwen2.5-VL and Ovis2) that have shown high ranks in the leaderboard on reasoning benchmarks.
> For the second experiment, as it is difficult to control and quantify how much of the axes are missed by VLM, we instead investigate the performance differences when we explicitly drop partial axes (10% and 20%) from the axes provided by VLMs. To this end, we first extract concept axes per image for entire images and only 80% or 90% of axes (i.e., 10%/20% random drop) are provided to our model in the training stage. We evaluate our method on the visual concept editing task in CelebA-HQ dataset.
>
> The below Table reveals that our method is hugely robust to both the choice of VLMs and missing axes. We conjecture that this is because even though some image-related axes are missed in each example, those axes will be eventually, repeatedly exposed across the dataset. Moreover, since our compositional anchoring loss encourages the compositionality of the concept representations, our framework might be internally trained for better compositional generalization. In fact, we observed that our method is capable of generating OOD samples (Figure 5 in our paper.)
>
> |  | Ours \+ InternVL2-5 | Ours \+ Qwen2.5-VL | Ours\+Ovis2 | Ours +InternVL2-5  \+ 10% drop  | Ours +InternVL2-5  \+ 20% drop  |
> |---|---|---|---|---|---|
> | CLIP Score ($\uparrow$) | 23.88 | 23.72 | 23.48 | 23.52 | 23.65 |
> | BLIP Score ($\uparrow$) | 49.58 | 48.64 | 48.35 | 48.69 | 48.61 |
>
> > Q4. More details on limitations, linked up to analyses of specific kinds of errors that the system makes, would be helpful.
>
>
> As the reviewer pointed, dependency on VLM can cause some errors in practice. For we found that VLM often wrongly annotates attributes of the small objects, or often confuses the attribute of close objects, which possibly affects the overall performance. We will add these specific failure cases in the Appendix.
>
>
> **Reference**
>
> [1], Lee et al., Language-informed visual concept learning, in ICLR 24.
>
>
> **Top-50 axes discovered by our framework**:
>
> [subject_type, background, color, material, shape, species, size, type, vehicle_type, environment, position, cap_color, gender, age, object_type, hair_color, hair_length, lighting, accessory, location, water_body, expression, clothing, activity, sky_condition, eye_color, body_color, windows, wing_shape, fur_texture, stem_color, surroundings, sail_color, seat_color, wing_color, wing_pattern, fruit_type, color_pattern, surface_texture, limbs, flower_color, style, design, wall_color, texture, brand, container_count, flooring, cap_texture, boat_color]

---

> ### Author Response · Authors · 2025-08-06
>
> Dear Reviewer vNjW,
>
> We would like to sincerely thank the reviewer again for their valuable comments. We hope that the additional experiments on a more diverse and unstructured dataset (a subset of ImageNet), along with our robustness analysis on VLMs, have addressed the reviewer’s concerns.
>
> We kindly follow up to inquire whether our response has sufficiently addressed the reviewer’s questions. Should the reviewer have any further queries or concerns, we would be happy to discuss.

---

### Note · Authors · 2025-08-13

We thank the reviewers and AC for their thorough evaluation and constructive feedback. We are pleased that three of the four reviewers have raised their scores, recognizing the paper’s enhanced applicability to complex data, robustness to performance of VLMs, and clear justification on key components of our framework.

**Positive assessments**

We appreciate the following positive assessments: well‑written and clearly motivated paper (vNjW, xMmD, 5Dzc); clear task definition of axis‑guided concept decomposition with modular, flexible design (5Dzc); sensible disentanglement framing with attribute‑editing capability (4ztc); consistent qualitative and quantitative gains over baselines (all reviewers); rich visualizations and a valuable human study (xMmD).

**Main concerns raised collectively by reviewers**

The main concerns were: (1) the need for broader evaluation on diverse, unstructured data (vNjW, 4ztc, 5Dzc); (2) robustness to the choice and performance of the VLM (all reviewers); and (3) ablations to justify key design choices (xMmD, 5Dzc).

**Our response**

We addressed these by: (1) showing consistent improvements over baselines on an ImageNet subset spanning animals, everyday objects, and scenes; (2) demonstrating robustness across three VLMs (InternVL2‑5, Qwen2.5‑VL, OVis2) and under random concept‑axis dropout up to 20%; and (3) performing ablations of the vision encoder (DINOv2 vs. CLIP), universal concept encoder (Q‑Former vs. shared MLP), and text‑to‑image model backbone (frozen vs. LoRA‑fine‑tuned), which justifies that our choices on model components lead to best performance.

**Remark**

We appreciate the sustained engagement of 4ztc, xMmD, and 5Dzc, and are glad to hear that our responses resolved their concerns. Regarding reviewer vNjW’s concerns, while we have not yet received acknowledgement of our rebuttal from the reviewer, we believe clarification on our work’s contribution, additional experiments on ImageNet subset, and robustness analysis on VLM address the key concerns, as the other reviewers acknowledged that those experiments properly addressed the same concerns. We will incorporate the additional experiments, ablations, and clarifications into the revised version of our paper. We appreciate the reviewers’ and AC’s efforts, which have substantially improved our paper, and hope these final remarks assist in decision‑making.

---

### Decision · Program_Chairs · 2025-09-17

**Decision:**

Accept (poster)

**Comment:**

The submission introduces a visual concept discovery framework for real-world images, which can be applied to compositional image editing and generation. The submission received two accept, one borderline accept, and one borderline reject ratings after rebuttal. Reviewer vNjW found the submission to not yet offer a fundamentally new approach guided by principles or theories, which the AC agrees. Overall, the AC believes the submission passes the acceptance bar for NeurIPS.